# Sequential Subset Matching for Dataset Distillation

**Jiawei Du , Qin Shi , Joey Tianyi Zhou**[✉]
Centre for Frontier AI Research (CFAR), Agency for Science, Technology and Research (A*STAR), Singapore
Institute of High Performance Computing (IHPC), Agency for Science, Technology and Research (A*STAR), Singapore
`{dujw,Joey_Zhou}@cfar.a-star.edu.sg`, shiqin924924@gmail.com

## Abstract

Dataset distillation is a newly emerging task that synthesizes a small-size dataset used in training deep neural networks (DNNs) for reducing data storage and model training costs. The synthetic datasets are expected to capture the essence of the knowledge contained in real-world datasets such that the former yields a similar performance as the latter. Recent advancements in distillation methods have produced notable improvements in generating synthetic datasets. However, current state-of-the-art methods treat the entire synthetic dataset as a unified entity and optimize each synthetic instance equally. This static optimization approach may lead to performance degradation in dataset distillation. Specifically, we argue that static optimization can give rise to a coupling issue within the synthetic data, particularly when a larger amount of synthetic data is being optimized. This coupling issue, in turn, leads to the failure of the distilled dataset to extract the high-level features learned by the deep neural network (DNN) in the latter epochs. In this study, we propose a new dataset distillation strategy called Sequential Subset Matching (SeqMatch), which tackles this problem by adaptively optimizing the synthetic data to encourage sequential acquisition of knowledge during dataset distillation. Our analysis indicates that SeqMatch effectively addresses the coupling issue by sequentially generating the synthetic instances, thereby enhancing its performance significantly. Our proposed SeqMatch outperforms state-of-the-art methods in various datasets, including SVNH, CIFAR-10, CIFAR-100, and Tiny ImageNet. Our code is available at https://github.com/shqii1j/seqmatch.

## 1 Introduction

Recent advancements in Deep Neural Networks (DNNs) have demonstrated their remarkable ability to extract knowledge from large-scale real-world data, as exemplified by the impressive performance of the large language model GPT-3, which was trained on a staggering 45 terabytes of text data [4]. However, the use of such massive datasets comes at a significant cost in terms of data storage, model training, and hyperparameter tuning.

The challenges associated with the use of large-scale datasets have motivated the development of various techniques aimed at reducing datasets size while preserving their essential characteristics. One such technique is dataset distillation [5, 6, 13, 22, 29, 35, 42, 48, 50, 51, 52, 53], which involves synthesizing a smaller dataset that effectively captures the knowledge contained within the original dataset. Models trained on these synthetic datasets have been shown to achieve comparable performance to those trained on the full dataset. In recent years, dataset distillation has garnered increasing attention from the deep learning community and has been leveraged in various practical applications, including continual learning [41, 52, 53], neural architecture search [21, 39, 40, 51, 53], and privacy-preserving tasks [12, 15, 31], among others.

---

[✉]Corresponding Author

37th Conference on Neural Information Processing Systems (NeurIPS 2023).

Existing methods for dataset distillation, as proposed in [5, 11, 13, 33, 37, 38, 47, 51, 53], have improved the distillation performance through enhanced optimization methods. These approaches have achieved commendable improvements in consolidating knowledge from the original dataset and generating superior synthetic datasets. However, the knowledge condensed by these existing methods primarily originates from the easy instances, which exhibit a rapid reduction in training loss during the early stages of training. These easy instances constitute the majority of the dataset and typically encompass low-level, yet commonly encountered visual features (e.g. edges and textures [49]) acquired in the initial epochs of training. In contrast, the remaining, less frequent, but more challenging instances encapsulate high-level features (e.g. shapes and contours) that are extracted in the subsequent epochs and significantly impact the generalization capability of deep neural networks (DNNs). The findings depicted in Figure 1 reveal that an overemphasis on low-level features hinders the extraction and condensation of high-level features from hard instances, thereby resulting in a decline in performance.

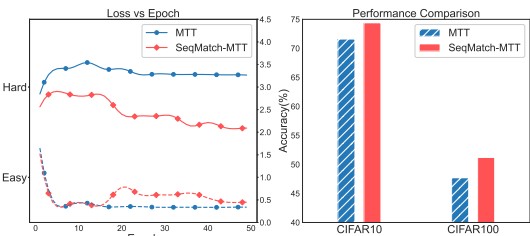

Figure 1: **Left:** MTT [5] fails to extract adequate high-level features. The loss drop rate between easy and hard instances is employed as the metric to evaluate the condensation efficacy of low-level and high-level features. The upper solid lines represent the loss change of hard instances, while the lower dashed lines depict the loss change of easy instances. The inability to decrease the loss of hard instances indicates MTT's inadequacy in capturing high-level features. In contrast, our proposed SeqMatch successfully minimizes the loss for both hard and easy instances. **Right:** The consequent performance improvement of SeqMatch in CIFAR [25] datasets. Experiments are conducted with 50 images per class (ipc = 50).

In this paper, we investigate the factors that hinder the efficient condensation of high-level features in dataset distillation. Firstly, we reveal that DNNs are optimized through a process of learning from low-level visual features and gradually adapting to higher-level features. The condensation of high-level features determines the effectiveness of dataset distillation. Secondly, we argue that existing dataset distillation methods fail to extract high-level features because they treat the synthetic data as a unified entity and optimize each synthetic instance unvaryingly. Such static optimization makes the synthetic instances become coupled with each other easier in cases where more synthetic instances are optimized. As a result, increasing the size of synthetic dataset will over-condense the low-level features but fail to condense additional knowledge from the real dataset, let alone the higher-level features.

Building upon the insights derived from our analysis, we present a novel dataset distillation strategy, termed Sequential Subset Matching (SeqMatch), which is designed to extract both low-level and high-level features from the real dataset, thereby improving dataset distillation. Our approach adopts a simple yet effective strategy for reorganizing the synthesized dataset $\mathcal{S}$ during the distillation and evaluation phases. Specifically, we divide the synthetic dataset into multiple subsets and encourage each subset to acquire knowledge in the order that DNNs learn from the real dataset. Our approach can be seamlessly integrated into existing dataset distillation methods. The experiments, as shown in Figure 1, demonstrate that SeqMatch effectively enables the latter subsets to capture high-level features. This, in turn, leads to a substantial improvement in performance compared to the baseline method MTT, which struggles to compress higher-level features from the real dataset. Extensive experiments demonstrate that SeqMatch outperforms state-of-the-art methods, particularly in high compression ratio[2] scenarios, across a range of datasets including CIFAR-10, CIFAR-100, TinyImageNet, and subsets of the ImageNet.

In a nutshell, our contribution can be summarized as follows.

- we examine the inefficacy of current dataset distillation in condensing hard instances from the original dataset. We present insightful analyses regarding the plausible factors contributing to this inefficacy and reveal the inherent preference of dataset distillation in condensing knowledge.

- We thereby propose a novel dataset distillation strategy called Sequential Subset Matching (SeqMatch) to targetedly encourage the condensing of higher-level features. SeqMatch seamlessly integrates with existing dataset distillation methods, offering easy implementa-

---

[2]compression ratio = compressed dataset size / full dataset size [8]

tion. Experiments on diverse datasets demonstrate the effectiveness of SeqMatch, achieving state-of-the-art performance.

## 2 Related work

Coreset selection is the traditional dataset reduction approach by selecting representative prototypes from the original dataset[2, 7, 17, 43, 45]. However, the non-editable nature of the coreset limits its performance potential. The idea of synthesizing the "coreset" can be traced back to Wang et al. [47]. Compared to coreset selection, dataset distillation has demonstrated greatly superior performance. Based on the approach of optimizing the synthetic data, dataset distillation can be taxonomized into two types: data-matching methods and meta-learning methods [30].

Data-matching methods encourage the synthetic data to imitate the influence of the target data, involving the gradients, trajectories, and distributions. Zhao and Bilen [52] proposed distribution matching to update synthetic data. Zhao et al. [53] matched the gradients of the target and synthetic data in each iteration for optimization. This approach led to the development of several advanced gradient-matching methods[5, 20, 22, 51]. Trajectory-matching methods [5, 9, 13] further matched multi-step gradients to optimize the synthetic data, achieving state-of-the-art performance. Factorization-based methods [11, 29, 33] distilled the synthetic data into a low-dimensional manifold and used a decoder to recover the source instances from the factorized features.

Meta-learning methods treat the synthetic data as the parameters to be optimized by a meta (or outer) algorithm [3, 11, 32, 34, 37, 38, 54]. A base (or inner) algorithm solves the supervised learning problem and is nested inside the meta (or outer) algorithm with respect to the synthetic data. The synthetic data can be directly updated to minimize the empirical risk of the network. Kernel ridge regression (KRR) based methods [34, 37, 38] have achieved remarkable performance among meta-learning methods.

Both data-matching and meta-learning methods optimize each synthetic instance equally. The absence of variation in converged synthetic instances may lead to the extraction of similar knowledge and result in over-representation of low-level features.

## 3 Preliminaries

**Background** Throughout this paper, we denote the target dataset as $\mathcal{T} = \{(x_i, y_i)\}_{i=1}^{|\mathcal{T}|}$. Each pair of data sample is drawn i.i.d. from a natural distribution $\mathcal{D}$, and $x_i \in \mathbb{R}^d, y_i \in \mathcal{Y} = \{0, 1, \cdots, C-1\}$ where $d$ is the dimension of input data and $C$ is the number of classes. We denote the synthetic dataset as $\mathcal{S} = \{(s_i, y_i)\}_{i=1}^{|\mathcal{S}|}$ where $s_i \in \mathbb{R}^d, y_i \in \mathcal{Y}$. Each class of $\mathcal{S}$ contains ipc (images per class) data pairs. Thus, $|\mathcal{S}| = \texttt{ipc} \times C$ and ipc is typically set to make $|\mathcal{S}| \ll |\mathcal{T}|$.

We employ $f_\theta$ to denote a deep neural network $f$ with weights $\theta$. An ideal training progress is to search for an optimal weight parameter $\hat{\theta}$ that minimizes the expected risk over the natural distribution $\mathcal{D}$, which is defined as $L_\mathcal{D}(f_\theta) \triangleq \mathbb{E}_{(x,y)\sim\mathcal{D}}\big[\ell(f_\theta(x), y)\big]$. However, as we can only access the training set $\mathcal{T}$ sampled from the natural distribution $\mathcal{D}$, the practical training approach of the network $f$ is to minimizing the empirical risk $L_\mathcal{T}(f_\theta)$ minimization (ERM) on the training set $\mathcal{T}$, which is defined as

$$\hat{\theta} = \texttt{alg}(\mathcal{T}, \theta_0) = \arg\min_\theta L_\mathcal{T}(f_\theta) \quad \text{where} \quad L_\mathcal{T}(f_\theta) = \frac{1}{|\mathcal{T}|} \sum_{x_i \in \mathcal{T}} \ell\big[f_\theta(x_i), y_i\big], \quad (1)$$

where $\ell$ can be any training loss function; alg is the given training algorithm that optimizes the initialized weights parameters $\theta_0$ over the training set $\mathcal{T}$; $\theta_0$ is initialized by sampling from a distribution $P_{\theta_0}$.

Dataset distillation aims to condense the knowledge of $\mathcal{T}$ into the synthetic dataset $\mathcal{S}$ so that training over the synthetic dataset $\mathcal{S}$ can achieve a comparable performance as training over the target dataset $\mathcal{T}$. The objective of dataset distillation can be formulated as,

$$\mathbb{E}_{(x,y)\sim\mathcal{D}, \theta_0\sim P_{\theta_0}} \big[\ell(f_{\texttt{alg}(\mathcal{T},\theta_0)}(x), y)\big] \quad \simeq \quad \mathbb{E}_{(x,y)\sim\mathcal{D}, \theta_0\sim P_{\theta_0}} \big[\ell(f_{\texttt{alg}(\mathcal{S},\theta_0)}(x), y)\big]. \quad (2)$$

**Gradient Matching Methods** We take gradient matching methods as the backbone method to present our distillation strategy. Matching the gradients introduced by $\mathcal{T}$ and $\mathcal{S}$ helps to solve $\mathcal{S}$

in Equation 2. By doing so, gradient matching methods achieve advanced performance in dataset distillation. Specifically, gradient matching methods introduce a distance metric $D(\cdot, \cdot)$ to measure the distance between gradients. A widely-used distance metric [53] is defined as $D(X, Y) = \sum_{i=1}^{I} \left( 1 - \frac{\langle X_i, Y_i \rangle}{\|X_i\| \|Y_i\|} \right)$, where $X, Y \in \mathbb{R}^{I \times J}$ and $X_i, Y_i \in \mathbb{R}^J$ are the $i^{\text{th}}$ columns of $X$ and $Y$ respectively. With the defined distance metric $D(\cdot, \cdot)$, gradient matching methods consider solving

$$\widehat{S} = \underset{\substack{S \subset \mathbb{R}^d \times \mathcal{Y} \\ |S| = \text{ipc} \times C}}{\arg\min} \underset{\theta_0 \sim P_{\theta_0}}{\mathbb{E}} \left[ \sum_{m=1}^{M} \mathcal{L}(S, \theta_m) \right], \quad \text{where} \quad \mathcal{L}(S, \theta) = D\big(\nabla_\theta L_S(f_\theta), \nabla_\theta L_\mathcal{T}(f_\theta)\big), \quad (3)$$

where $\theta_i$ is the intermediate weights which is continuously updated by training the network $f_{\theta_0}$ over the target dataset $\mathcal{T}$. The methods employ $M$ as the hyperparameter to control the length of teacher trajectories to be matched starting from the initialized weights $\theta_0 \sim P_{\theta_0}$. $\mathcal{L}(S.\theta)$ is the matching loss. The teacher trajectory $\{\theta_0, \theta_1, \cdots, \theta_M\}$ is equivalent to a series of gradients $\{g_1, g_2, \cdots, g_M\}$. To ensure the robustness of the synthetic dataset $S$ to different weights initializations, $\theta_0$ will be sampled from $P_{\theta_0}$ for many times. As a consequence, the distributions of the gradients for training can be represented as $\{P_{g_1}, P_{g_2}, \cdots, P_{g_M}\}$.

---

**Algorithm 1** Training with SeqMatch in Distillation Phase.

---

**Input:** Target dataset $\mathcal{T}$; Number of subsets $K$; Iterations $N$ in updating each subset; A base distillation method $\mathcal{A}$.

1: Initialize the synthetic dataset $S_{\text{all}}$
2: Divide $S_{\text{all}}$ into $K$ subsets of equal size $\lfloor \frac{|S_{\text{all}}|}{K} \rfloor$, i.e., $S_{\text{all}} = S_1 \cup S_2 \cup \cdots \cup S_K$
3: **for each** $S_k$ **do**
4:     $\triangleright$ Optimize each subset $S_k$ sequentially:
5:     **repeat**
6:         **if** k =1 **then**
7:             Initialize network weights $\theta_0^k \sim P_{\theta_0}$
8:         **else**
9:             Load network weights $\theta_0^k \sim P_{\theta_N^{k-1}}$ saved in optimizing last subset $S_{k-1}$
10:         **for** $i = 1$ to $N$ **do**
11:             $\triangleright$ Update Network weights by subset $S_k$:
12:             $\theta_i^k = \text{alg}(S_k \cup \mathbb{S}^{(k-1)}, \theta_{i-1}^k)$
13:             $\triangleright$ Update $S_k$ by the base distillation method:
14:             $S_k \leftarrow \mathcal{A}(\mathcal{T}, S_k, \theta_i^k)$
15:         Record and save updated network weights $\theta_N^{k-1}$
16:     **until** Converge

**Output:** Distilled synthetic dataset $S_{\text{all}}$

---

# 4 Method

Increasing the size of a synthetic dataset is a straightforward approach to incorporating additional high-level features. However, our findings reveal that simply optimizing more synthetic data leads to an excessive focus on knowledge learned from easy instances. In this section, we first introduce the concept of sequential knowledge acquisition in a standard training procedure (refer to subsection 4.1). Subsequently, we argue that the varying rate of convergence causes certain portions of the synthetic data to abandon the extraction of further knowledge in the later stages (as discussed in Figure 4.2). Finally, we present our proposed strategy, Sequential Subset Matching (referred to as SeqMatch), which is outlined in Algorithm 1 in subsection 4.3.

## 4.1 Features Are Represented Sequentially

Many studies have observed the sequential acquisition of knowledge in training DNNs. Zeiler et al. [49] revealed that DNNs are optimized to extract low-level visual features, such as edges and textures, in the lower layers, while higher-level features, such as object parts and shapes, were

represented in the higher layers. Han et al. [16] leverages the observation that DNNs learn the knowledge from easy instances first, and gradually adapt to hard instances[1] to propose noisy learning methods. The sequential acquisition of knowledge is a critical aspect of DNN.

However, effectively condensing knowledge throughout the entire training process presents significant challenges for existing dataset distillation methods. While the synthetic dataset $\mathcal{S}$ is employed to learn from extensive teacher trajectories, extending the length of these trajectories during distillation can exacerbate the issue of domain shifting in gradient distributions, thereby resulting in performance degradation. This is primarily due to the fact that the knowledge extracted from the target dataset $\mathcal{T}$ varies across different epochs, leading to corresponding shifts in the domains of gradient distributions. Consequently, the synthetic dataset $\mathcal{S}$ may struggle to adequately capture and consolidate knowledge from prolonged teacher trajectories.

To enhance distillation performance, a common approach is to match a shorter teacher trajectory while disregarding knowledge extracted from the latter epochs of $\mathcal{T}$. For instance, in the case of the CIFAR-10 dataset, optimal hyperparameters for $M$ (measured in epochs) in the MTT [5] method were found to be $2, 20, 40$ for $\texttt{ipc} = 1, 10, 50$ settings, respectively. The compromise made in matching a shorter teacher trajectory unexpectedly resulted in a performance gain, thereby confirming the presence of excessive condensation on easy instances.

Taking into account the sequential acquisition of knowledge during deep neural network (DNN) training is crucial for improving the generalization ability of synthetic data. Involving more synthetic data is the most straightforward approach to condense additional knowledge from longer teacher trajectory. However, our experimental findings, as illustrated in Figure 1, indicate that current gradient matching methods tend to prioritize the consolidation of knowledge derived from easy instances in the early epochs. Consequently, we conducted further investigations into the excessive emphasis on low-level features in existing dataset distillation methods.

## 4.2    Coupled Synthetic Dataset

The coupling issue within the synthetic dataset impedes its effectiveness in condensing additional high-level features. Existing dataset distillation methods optimize the synthetic dataset $\mathcal{S}$ as a unified entity, resulting in the backpropagated gradients used to update $\mathcal{S}$ being applied globally. The gradients on each instance only differ across different initializations and pre-assigned labels, implying that instances sharing a similar initialization within the same class will converge similarly. Consequently, a portion of the synthetic data only serves the purpose of alleviating the gradient matching error for the pre-existing synthetic data.

Consider a synthetic dataset $\mathcal{S}$ is newly initialized to be distilled from a target dataset $\mathcal{T}$. The distributions of the gradients for distillation are $\{P_{g_1}, P_{g_2}, \cdots, P_{g_M}\}$, and the sampled gradients for training is $\{g_1, g_2, \cdots, g_M\}$. Suppose that $G$ is the integrated gradients calculated by $\mathcal{S}$, by minimizing the loss function as stated in Equation 3, the gradients used for updating $s_i$ when $\theta = \theta_m$ would be

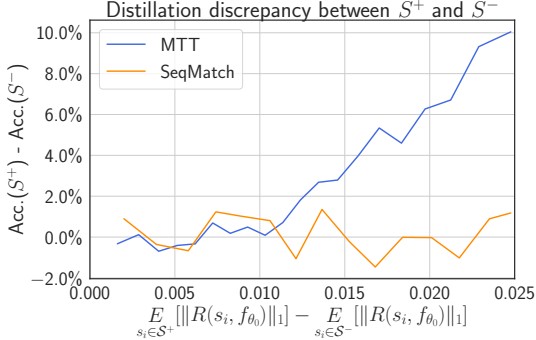

Figure 2: The accuracy discrepancy between the networks trained using $\mathcal{S}^+$ and $\mathcal{S}^-$ separately. The discrepancy will increase with the magnitude of $R(s_i, f_{\theta_m})$. These results verified the coupling issue between $\mathcal{S}^+$ and $\mathcal{S}^-$, and our proposed method SeqMatch successfully mitigates the coupling issue. More experimental details can be found in subsection 5.3.

$$
\begin{aligned}
\nabla_{s_i} \mathcal{L}(\mathcal{S}, \theta_m) &= \frac{\partial \mathcal{L}}{\partial G} \cdot \frac{\partial G}{\partial \nabla_{\theta_m} \ell(f_{\theta_m}(s_i), y_i)} \cdot \frac{\partial \nabla_{\theta_m} \ell(f_{\theta_m}(s_i), y_i)}{\partial s_i} \\
&= \frac{\partial \mathcal{L}}{\partial G} \cdot R(s_i, f_{\theta_m}), \quad \text{where} \quad R(s_i, f_{\theta_m}) \triangleq \frac{\partial \nabla_{\theta_m} \ell(f_{\theta_m}(s_i), y_i)}{\partial s_i}.
\end{aligned} \quad (4)
$$

we have $\frac{\partial G}{\partial \nabla_{\theta_m} \ell(f_{\theta_m}(s_i), y_i)} = 1$, because $G$ is accumulated by the gradients of each synthetic data, i.e., $G = \nabla_{\theta_m} L_{\mathcal{S}}(f_{\theta_m}) = \sum_{i=1}^{\|\mathcal{S}\|} \nabla_{\theta_m} \ell(f_{\theta_m}(s_i), y_i)$. Here we define the amplification function $R(s_i, f_{\theta_m}) \subset \mathbb{R}^d$. Then, the gradients on updating synthetic instance $\nabla_{s_i} \mathcal{L}(\mathcal{S}, \theta_m)$ shares the same $\frac{\partial \mathcal{L}}{\partial G}$ and only varies in $R(s_i, f_{\theta_m})$. The amplification function $R(s_i, f_{\theta_m})$ is only affected by the pre-assigned label and initialization of $s_i$.

More importantly, the magnitude of $R(s_i, f_{\theta_m})$ determines the rate of convergence of each synthetic instance $s_i$. Sorted by the $l_1$-norm of amplification function $\|R(s_i, f_{\theta_m})\|_1$ can be divided into two subsets $\mathcal{S}^+$ and $\mathcal{S}^-$. $\mathcal{S}^+$ contains the synthetic instances with greater values of $R(s_i, f_{\theta_m})$ than those in $\mathcal{S}^-$. That implies that instances in $\mathcal{S}^+$ converge faster to minimize $D(\nabla_{\theta_m} L_{\mathcal{S}^+}(f_{\theta_m}), g_m)$, and $S^+$ is optimized to imitate $g_m$. On the other hand, the instances in $\mathcal{S}^-$ converge slower and are optimized to minimize $D(\nabla_{\theta_m} L_{\mathcal{S}^-}(f_{\theta_m}), \epsilon)$, where $\epsilon$ represents the gradients matching error $\epsilon$ of $S^+$, i.e., $\epsilon = g_m - \nabla_{\theta_m} L_{\mathcal{S}^+}(f_{\theta_m})$. Therefore, $S^-$ is optimized to imitate $\epsilon$ and its effectiveness is achieved by compensating for the gradients matching error of $S^+$. $S^-$ is coupled with $S^+$ and unable to capture the higher-level features in the latter epochs.

We conducted experiments to investigate whether $S^-$ solely compensates for the gradient matching error of $S^+$ and is unable to extract knowledge independently. To achieve this, we sorted $\mathcal{S}^+$ and $\mathcal{S}^-$ by the $l_1$-norm of the amplification function $\|R(s_i, f_{\theta_m})\|_1$ and trained separate networks with $\mathcal{S}^+$ and $\mathcal{S}^-$. As depicted in Figure 2, we observed a significant discrepancy in accuracy, which increased with the difference in magnitude of $R(s_i, f_{\theta_m})$. Further details and discussion are provided in subsection 5.3. These experiments verify the coupling issue wherein $S^-$ compensates for the matching error of $S^+$, thereby reducing its effectiveness in condensing additional knowledge.

## 4.3 Sequential Subset Matching

We can use a standard deep learning task as an analogy for the dataset distillation problem, then the synthetic dataset $\mathcal{S}$ can be thought of as the weight parameters that need to be optimized. However, simply increasing the size of the synthetic dataset is comparable to multiplying the parameters of a model in an exact layer without architecting the newly added parameters, and the resulting performance improvement is marginal. We thereby propose SeqMatch to reorganize the synthetic dataset $\mathcal{S}$ to utilize the newly added synthetic data.

We incorporate additional variability into the optimization process of synthetic data to encourage the capture of higher-level feature extracted in the latter training progress. To do this, SeqMatch divides the synthetic dataset $\mathcal{S}$ into $K$ subsets equally, i.e., $\mathcal{S} = \mathcal{S}_1 \cup \mathcal{S}_2 \cup \cdots \cup \mathcal{S}_K, |\mathcal{S}_k| = \lfloor \frac{|\mathcal{S}|}{K} \rceil$. SeqMatch optimizes each $\mathcal{S}_k$ by solving

$$\widehat{\mathcal{S}}_k = \underset{\substack{\mathcal{S}_k \subset \mathbb{R}^d \times \mathcal{Y} \\ |\mathcal{S}_k| = \lfloor |\mathcal{S}|/K \rceil}}{\arg\min} \underset{\theta_0 \sim P_{\theta_0}}{\mathbb{E}} \left[ \sum_{m=(k-1)n}^{kn} \mathcal{L}(\mathcal{S}_k \cup \mathbb{S}^{(k-1)}, \theta_m) \right], \tag{5}$$

where $\mathbb{S}^{(k-1)} = \mathcal{S}_1 \cup \mathcal{S}_2 \cup \cdots \cup \mathcal{S}_{k-1}$, which represents the union set of the subsets in the former. $\mathbb{S}^{(k-1)}$ are fixed and only $\mathcal{S}_k$ will be updated. The subset $\mathcal{S}_k$ is encouraged to match the corresponding $k^{th}$ segment of the teacher trajectory to condense the knowledge in the latter epoch. Let $n = \lfloor \frac{M}{K} \rceil$ denote the length of trajectory segment to be matched by each subset $\mathcal{S}_k$ in the proposed framework. To strike a balance between providing adequate capacity for distillation and avoiding coupled synthetic data , the size of each subset $\mathcal{S}_k$ is well-controlled by $K$.

In the distillation phase, each subset is arranged in ascending order to be optimized sequentially. We reveal that the first subset $\mathcal{S}_1$ with $\frac{1}{K}$ size of the original synthetic dataset $\mathcal{S}$ is sufficient to condense adequate knowledge in the former epoch. For the subsequent subset $\mathcal{S}_k$, we encourage the $k^{th}$ subset $\mathcal{S}_k$ condense the knowledge different from those condensed in the previous subsets. This is achieved by minimizing the matching loss $\mathcal{L}(\mathcal{S}_k \cup \mathbb{S}^{(k-1)}, \theta_m)$ while only $\mathcal{S}_k$ will be updated.

During the evaluation phase, the subsets of the synthetic dataset are used sequentially to train the neural network $f_\theta$, with the weight parameters $\theta$ being iteratively updated by $\theta_k = \text{alg}(\mathcal{S}_k, \theta_{k-1})$. This training process emulates the sequential feature extraction of real dataset $\mathcal{T}$ during training. Further details regarding SeqMatch and the optimization of $\theta^*$ can be found in Algorithm.1.

# 5  Experiment

In this section, we provide implementation details for our proposed method, along with instructions for reproducibility. We compare the performance of SeqMatch against state-of-the-art dataset distillation methods on a variety of datasets. To ensure a fair and comprehensive comparison, we follow up the experimental setup as stated in [8, 30]. We provide more experiments to verify the effectiveness of SeqMatch including the results on ImageNet subsets and analysis experiments in Appendix A.1 due to page constraints.

## 5.1  Experimental Setup

**Datasets:** We evaluate the performance of dataset distillation methods on several widely-used datasets across various resolutions. MNIST [28], which is a fundamental classification dataset, is included with a resolution of $28 \times 28$. SVNH [36] is also considered, which is composed of RGB images of house numbers cwith a resolution of $32 \times 32$. CIFAR10 and CIFAR100 [25], two datasets frequently used in dataset distillation, are evaluated in this study. These datasets consist of $50,000$ training images and $10,000$ test images from 10 and 100 different categories, respectively. Additionally, our proposed method is evaluated on the Tiny ImageNet [27] dataset with a resolution of $64 \times 64$ and on the ImageNet [24] subsets with a resolution of $128 \times 128$.

**Evaluation Metric:** The evaluation metric involves distillation phase and evaluation phase. In the former, the synthetic dataset is optimized with a distillation budget that typically restricts the number of images per class (`ipc`). We evaluate the performance of our method and baseline methods under the settings `ipc` $= \{10, 50\}$. We do not evaluate the setting with `ipc` $= 1$ since our approach requires `ipc` $\geq 2$ . To facilitate a clear comparison, we mark the factorization-based baselines with an asterisk (*) since they often employ an additional decoder, following the suggestion in [30]. We employ 4-layer ConvNet [14] in Tiny ImageNet dataset whereas for the other datasets we use a 3-layer ConvNet [14].

In the evaluation phase, we utilize the optimized synthetic dataset to train neural networks using a standard training procedure. Specifically, we use each synthetic dataset to train five networks with random initializations for $1,000$ iterations and report the mean accuracy and its standard deviation of the results.

**Implementation Details.** To ensure the reproducibility of SeqMatch, we provide detailed implementation specifications. Our method relies on a single hyperparameter, denoted by $K$, which determines the number of subsets. In order to balance the inclusion of sufficient knowledge in each segment with the capture of high-level features in the later stages, we set $K = \{2, 3\}$ for the scenarios where `ipc` $= \{10, 50\}$, respectively. Notably, our evaluation results demonstrate that the choice of $K$ remains consistent across the various datasets.

As a plug-in strategy, SeqMatch requires a backbone method for dataset synthesis. Each synthetic subset is optimized using a standard training procedure, specific to the chosen backbone method. The only hyperparameters that require adjustment in the backbone method are those that control the segments of the teacher trajectory to be learned by the synthetic dataset, whereas the remaining hyperparameters emain consistent without adjustment. Such adjustment is to ensure each synthetic subset effectively condenses the knowledge into stages. The precise hyperparameters of the backbone methods are presented in Appendix A.3. We conduct our experiments on the server with four Tesla V100 GPUs.

## 5.2  Results

Our proposed SeqMatch is plugged into the methods MTT [5] and IDC [23], which are denoted as SeqMatch-MTT and SeqMatch-IDC, respectively. As shown in Table 1, the classification accuracies of ConvNet [14] trained using each dataset distillation method are summarized. The results indicate that SeqMatch significantly outperforms the backbone method across various datasets, and even surpasses state-of-the-art baseline methods in different settings. Our method is demonstrated to outperform the state-of-the-art baseline methods in different settings among different datasets. Notably, SeqMatch achieves a greater performance improvement in scenarios with a high compression ratio (i.e., `ipc` $= 50$). For instance, we observe a $3.5\%$ boost in the performance of MTT [5], achieving $51.2\%$ accuracy on CIFAR-100. Similarly, we observe a $1.9\%$ performance enhancement in IDC [23],

Table 1: Performance comparison of dataset distillation methods across a variety of datasets. Abbreviations of GM, TM, DM,META stand for gradient matching, trajectory matching, distribution matching, and meta-learning respectively. We reproduce the results of MTT [5] and IDC [23] and cite the results of the other baselines [30]. The best results of non-factorized methods (without decoders) are highlighted in orange font. The best results of factorization-based methods are highlighted in blue font.

| Methods | Schemes | MNIST | | SVHN | | CIFAR-10 | | CIFAR-100 | | Tiny ImageNet |
|---|---|---|---|---|---|---|---|---|---|---|
| | | 10 | 50 | 10 | 50 | 10 | 50 | 10 | 50 | 10 |
| DD [47] | META | 79.5 ±8.1 | - | - | - | 36.8 ±1.2 | - | - | - | - |
| DC [53] | GM | 94.7 ±0.2 | 98.8 ±0.2 | 76.1 ±0.6 | 82.3 ±0.3 | 44.9 ±0.5 | 53.9 ±0.5 | 32.3 ±0.3 | 42.8 ±0.4 | - |
| DSA [51] | GM | 97.8 ±0.1 | 99.2 ±0.1 | 79.2 ±0.5 | 84.4 ±0.4 | 52.1 ±0.5 | 60.6 ±0.5 | 32.3 ±0.3 | 42.8 ±0.4 | - |
| DM [52] | DM | 97.3 ±0.3 | 94.8 ±0.2 | - | - | 48.9 ±0.6 | 63.0 ±0.4 | 29.7 ±0.3 | 43.6 ±0.4 | 12.9 ±0.4 |
| CAFE [46] | DM | 97.5 ±0.1 | 98.9 ±0.2 | 77.9 ±0.6 | 82.3 ±0.4 | 50.9 ±0.5 | 62.3 ±0.4 | 31.5 ±0.2 | 42.9 ±0.2 | - |
| KIP [37, 38] | KRR | 97.5 ±0.0 | 98.3 ±0.1 | 75.0 ±0.1 | 85.0 ±0.1 | 62.7 ±0.3 | 68.6 ±0.2 | 28.3 ±0.1 | - | - |
| FTD [13] | TM | - | - | - | - | 66.6 ±0.3 | 73.8 ±0.2 | 43.4 ±0.3 | 50.7 ±0.3 | 24.5 ±0.2 |
| MTT [5] | TM | 97.3 ±0.1 | 98.5 ±0.1 | 79.9 ± | 87.7 ±0.3 | 65.3 ±0.7 | 71.6 ±0.2 | 40.1 ±0.4 | 47.7 ±0.2 | 23.2 ±1.3 |
| SeqMatch-MTT | TM | 97.6 ±0.2 | 99.0 ±0.1 | 80.2 ±0.6 | 88.5 ±0.2 | 66.2 ±0.6 | 74.4 ±0.5 | 41.9 ±0.3 | 51.2 ±0.3 | 23.8 ±0.3 |
| IDC* [23] [3] | GM | 98.4 ±0.1 | 99.1 ±0.1 | 87.3 ±0.2 | 90.2 ±0.1 | 67.5 ±0.5 | 74.5 ±0.1 | 44.8 ±0.2 | 51.4 ±0.4 | - |
| SeqMatch-IDC* | GM | 98.6 ±0.1 | 99.2 ±0.0 | 87.2 ±0.2 | 92.1 ±0.1 | 68.3 ±0.2 | 75.3 ±0.2 | 45.1 ±0.3 | 51.9 ±0.3 | - |
| RTP* [11] | META | 99.3 ±0.5 | 99.4 ±0.4 | 89.1 ±0.2 | 89.5 ±0.2 | 71.2 ±0.4 | 73.6 ±0.4 | 42.9 ±0.7 | - | - |
| HaBa* [33] | TM | - | - | 83.2 ±0.4 | 88.3 ±0.1 | 69.9 ±0.4 | 74.0 ±0.2 | 40.2 ±0.2 | 47.0 ±0.2 | - |
| Whole | - | 99.6 ±0.0 | | 95.4 ±0.1 | | 84.8 ±0.1 | | 56.2 ±0.3 | | 37.6 ±0.4 |

achieving 92.1% accuracy on SVNH, which approaches the 95.4% accuracy obtained using the real dataset. These results suggest that our method is effective in mitigating the adverse effects of coupling and effectively condenses high-level features in high compression ratio scenarios.

**Cross-Architecture Generalization:** We also conducted experiments to evaluate cross-architecture generalization, as illustrated in Table 2. The ability to generalize effectively across different architectures is crucial for the practical application of dataset distillation. We evaluated our proposed SeqMatch on the CIFAR-10 dataset with ipc = 50. Following the evaluation metric established in [13, 46], three additional neural network architectures were utilized for evaluation: ResNet [19], VGG [44], and AlexNet [26]. Our SeqMatch approach demonstrated a significant improvement in performance during cross-architecture evaluation, highlighting its superior generalization ability.

## 5.3 Discussions

**Sequential Knowledge Acquisition:** We conducted experiments on CIFAR-10 with ipc = 50, presented in Figure 1, to investigate the inability of existing baseline methods to capture the knowledge learned in the latter epoch, as discussed in subsection 4.1. Inspired by [16], we utilized the change in instance-wise loss on real dataset to measure the effectiveness of condensing high-level features. Specifically, we recorded the loss of each instance from the real dataset $\mathcal{T}$ at every epoch, where the network was trained with synthetic dataset for only 20

Table 2: Cross-Architecture Results trained with ConvNet on CIFAR-10 with ipc = 50. We cite the results reported in Du et al. [13].

| Method | Evaluation Model | | | |
|---|---|---|---|---|
| | ConvNet | ResNet18 | VGG11 | AlexNet |
| DC [53] | 53.9 ±0.5 | 20.8 ±1.0 | 38.8 ±1.1 | 28.7 ±0.7 |
| CAFE [46] | 55.5 ±0.4 | 25.3 ±0.9 | 40.5 ±0.8 | 34.0 ±0.6 |
| MTT [5] | 71.6 ±0.2 | 61.9 ±0.7 | 55.4 ±0.8 | 48.2 ±1.0 |
| FTD [13] | 73.8 ±0.2 | 65.7 ±0.3 | 58.4 ±1.6 | 53.8 ±0.9 |
| SeqMatch-MTT | 74.4 ±0.5 | 68.4 ±0.9 | 64.2 ±0.7 | 50.7 ±1.0 |
| SeqMatch-IDC | 75.3 ±0.2 | 69.7 ±0.6 | 73.4 ±0.1 | 72.0 ±0.2 |

---

[3]Although IDC [23] is not categorized as a factorization-based method, it employs data parameterization to better improve the performance of synthetic dataset. Therefore, we compare IDC to the factorization-based method as factorization can be treated as a kind of special data parameterization.

iterations in each epoch. To distinguish hard instances from easy ones, we employed k-means algorithm [18] to cluster all instances in the real dataset into two clusters based on the recorded instance-wise loss. The distribution of instances in terms of difficulty is as follows: 77% are considered easy instances, while 23% are classified as hard instances.

We evaluated MTT[5] and SeqMatch as mentioned above. Our results show that MTT[5] over-condenses the knowledge learned in the former epoch. In contrast, SeqMatch is able to successfully capture the knowledge learned in the latter epoch.

**Coupled Synthetic Subsets:** In order to validate our hypothesis that the synthetic subset $\mathcal{S}^-$ is ineffective at condensing knowledge independently and results in over-condensation on the knowledge learned in the former epoch, we conducted experiments as shown in Figure 2. We sorted the subsets $\mathcal{S}^+$ and $\mathcal{S}^-$ of the same size by the $l_1$-norm of the amplification function $|R(s_i, f_{\theta_m})|1$ as explained in Figure 4.2. We then recorded the accuracy discrepancies between the separate networks trained by $\mathcal{S}^+$ and $\mathcal{S}^-$ with respect to the mean $l_1$-norm difference, i.e., $\mathbb{E}_{s_i \in \mathcal{S}^+}[|R(s_i, f_{\theta_0})|1] - \mathbb{E}_{s_i \in \mathcal{S}^-}[|R(s_i, f_{\theta_0})|_1]$.

As shown in Figure 2, the accuracy discrepancies increased linearly with the $l_1$-norm difference, which verifies our hypothesis that $\mathcal{S}^-$ is coupled with $\mathcal{S}^+$ and this coupling leads to the excessive condensation on low-level features. However, our proposed method, SeqMatch, is able to alleviate the coupling issue by encouraging $\mathcal{S}^-$ to condense knowledge more efficiently.

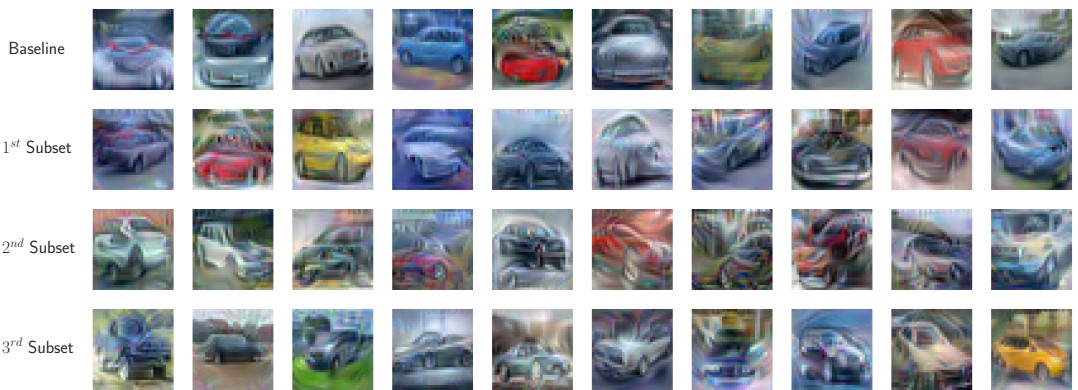

Figure 3: Visualization example of "car" synthetic images distilled by MTT [5] and SeqMatch from $32 \times 32$ CIFAR-10 (`ipc` = 50).

**Synthetic Image Visualization:** In order to demonstrate the distinction between MTT [5] and SeqMatch, we visualized synthetic images within the "car" class from CIFAR-10 [25] and visually compared them. As depicted in Figure 3, the synthetic images produced by MTT exhibit more concrete features and closely resemble actual "car" images. Conversely, the synthetic images generated by SeqMatch in the $2^{nd}$ and $3^{rd}$ subsets possess more abstract attributes and contain complex car shapes. We provide more visualizations of the synthetic images in Appendix A.2.

## 5.4 Limitations and Future Work

We acknowledge the limitations of our work from two perspectives. Firstly, our proposed sequential optimization of synthetic subsets increases the overall training time, potentially doubling or tripling it. To address this, future research could investigate optimization methods that allow for parallel optimization of each synthetic subset. Secondly, as the performance of subsequent synthetic subsets builds upon the performance of previous subsets, a strategy is required to adaptively distribute the distillation budget of each subset. Further research could explore strategies to address this limitation and effectively enhance the performance of dataset distillation, particularly in high compression ratio scenarios.

## 6  Conclusion

In this study, we provide empirical evidence of the failure in condensing high-level features in dataset distillation attributed to the sequential acquisition of knowledge in training DNNs. We reveal that the static optimization of synthetic data leads to a bias in over-condensing the low-level features, predominantly extracted from the majority during the initial stages of training. To address this issue in a targeted manner, we introduce an adaptive and plug-in distillation strategy called SeqMatch. Our proposed strategy involves the division of synthetic data into multiple subsets, which are sequentially optimized, thereby promoting the effective condensation of high-level features learned in the later epochs. Through comprehensive experimentation on diverse datasets, we validate the effectiveness of our analysis and proposed strategy, achieving state-of-the-art performance.

## Acknowledgements

This work is support by Joey Tianyi Zhou's A*STAR SERC Central Research Fund (Use-inspired Basic Research) and the Singapore Government's Research, Innovation and Enterprise 2020 Plan (Advanced Manufacturing and Engineering domain) under Grant A18A1b0045.

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

# A   More Experiements

## A.1   ImageNet Subsets

To assess the efficacy of our approach, we conducted experiments on subsets of the ImageNet dataset [10]. These subsets were constructed by selecting ten pertinent categories from the ImageNet-1k dataset [10], with a resolution of $128 \times 128$. Consequently, the ImageNet subsets pose greater challenges compared to the CIFAR-10/100 [25] and Tiny ImageNet [27] datasets. We adhered to the configuration of the ImageNet subsets as suggested by previous studies [5, 13, 33], encompassing subsets such as ImageNette (diverse objects), ImageWoof (dog breeds), ImageFruits (various fruits), and ImageMeow (cats).

To synthesize the dataset, we employed a 5-layer ConvNet [14] with a parameter setting of `ipc` $= 10$. The evaluation of the synthetic dataset involved performing five trials with randomly initialized networks. We compared the outcomes of our proposed method, referred to as SeqMatch, with the baseline approach MTT [5], as well as the plug-in strategies FTD [13] and HaBa [33] which build upon MTT.

The comprehensive results are presented in Table 3. Our proposed SeqMatch consistently outperformed the baseline MTT across all subsets. Notably, we achieved a performance improvement of $4.3\%$ on the ImageFruit subset. Additionally, SeqMatch demonstrated superior performance compared to HaBa and achieved comparable results to FTD.

Table 3: The performance comparison trained with 5-layer ConvNet on the ImageNet subsets with a resolution of $128 \times 128$. We cite the results as reported in MTT [5], FTD [13] and HaBa [33]. The latter two methods, FTD and HaBa, are plug-in strategies that build upon the foundation of MTT. Our proposed approach, SeqMatch, exhibits superior performance compared to both MTT and HaBa, demonstrating a significant improvement in results.

|  | ImageNette | ImageWoof | ImageFruit | ImageMeow |
|---|---|---|---|---|
| Real dataset | 87.4 ±1.0 | 67.0 ±1.3 | 63.9 ±2.0 | 66.7 ±1.1 |
| MTT [5] | 63.0 ±1.3 | 35.8 ±1.8 | 40.3 ±1.3 | 40.4 ±2.2 |
| FTD [13] | 67.7 ±1.3 | 38.8 ±1.4 | 44.9 ±1.5 | 43.3 ±0.6 |
| HaBa* [33] | 64.7 ±1.6 | 38.6 ±1.2 | 42.5 ±1.6 | 42.9 ±0.9 |
| SeqMatch-MTT | 66.9 ±1.7 | 38.7 ±1.1 | 44.6 ±1.7 | 44.8 ±1.2 |
| SeqMatch-FTD | **70.6** ±1.5 | **41.1** ±1.4 | **46.5** ±1.2 | **45.4** ±1.2 |

## A.2   More Visualizations

**Instance-wise loss change**: We have presented the average loss change of easy and hard instances in Figure 1, revealing that MTT failed to effectively condense the knowledge learned from the hard instances. To avoid the bias introduced by averaging, we have meticulously recorded and visualized the precise loss change of each individual instance. This is accomplished by employing a heatmap representation, as demonstrated in Figure 6. Each instance is depicted as a horizontal line exhibiting varying colors, with deeper shades of blue indicating higher loss values. Following the same clustering approach as depicted in Figure 1, we proceed to visualize the hard instances at the top and the easy instances at the bottom.

The individual loss changes of MTT, depicted in Figure 6, remain static across epochs. The losses of easy instances decrease to a small value during the initial stages, while the losses of hard instances persist at a high value until the end of training. These results confirm that MTT excessively focuses on low-level features. In contrast, the visualization of SeqMatch clearly exhibits the effect of color gradient, indicating a decrease in loss for the majority of instances. Notably, the losses of hard instances experience a significant decrease when a subsequent subset is introduced. These results validate that SeqMatch effectively consolidates knowledge in a sequential manner.

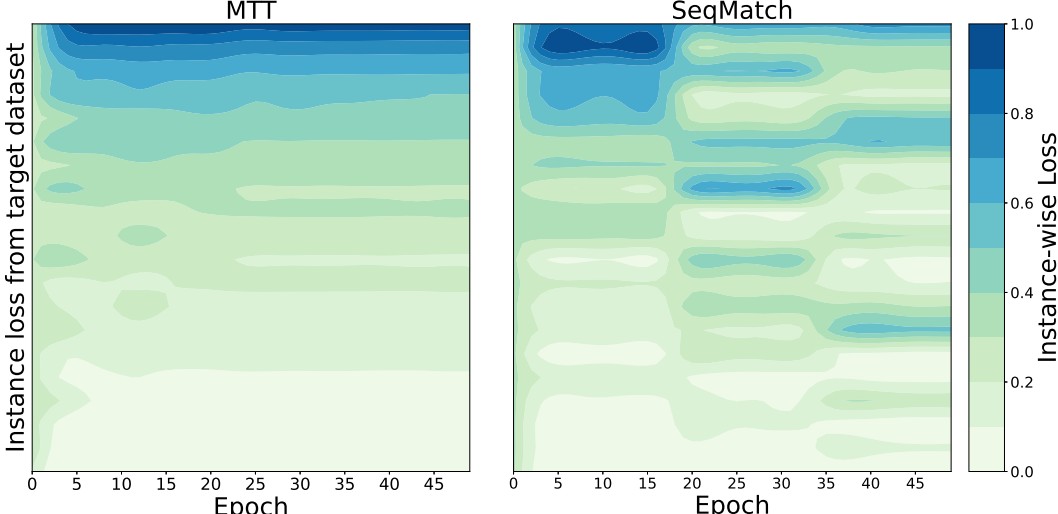

Figure 4: The heatmap illustrates the loss change of each instance in the real dataset across epochs. Each row in the heatmap represents an instance, while the deeper blue color denotes higher instance-wise loss. The network is trained with the synthetic datasets distilled by MTT and SeqMatch. **Left**: MTT fails to reduce the loss of hard instances while excessively reducing the loss of easy instances. **Right**: SeqMatch minimizes the loss of both the hard and easy instances.

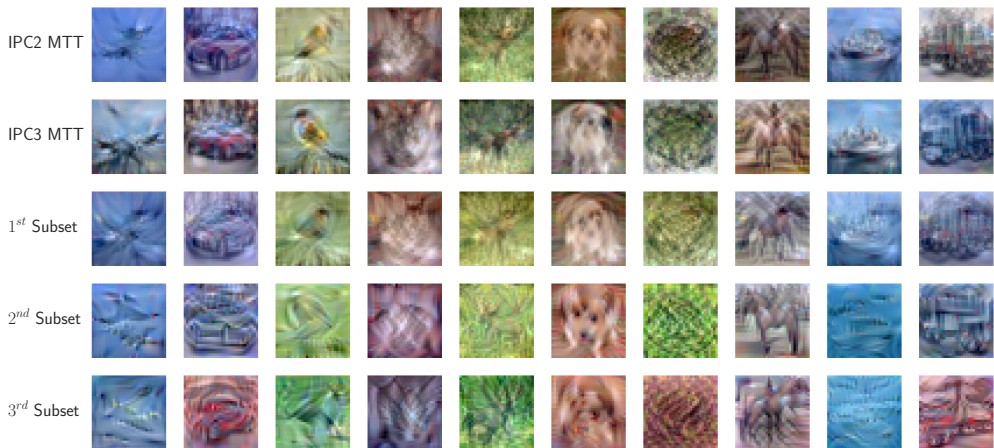

Figure 5: Visualization example of synthetic images distilled by MTT [5] and SeqMatch from $32 \times 32$ CIFAR-10 ($\texttt{ipc} = \{2, 3\}$). SeqMatch(ipc=2) is seamlessly embedded as the first two rows within the nameSeqMatch(ipc=3) visualization.

**Synthetic Dataset Visualization**: We compare the synthetic images with $\texttt{ipc} = \{2, 3\}$ from the CIFAR10 dataset to highlight the differences between subsets of Seqmatch in Figure 5. We provide more visualizations of the synthetic datasets for $\texttt{ipc} = 10$ from the $128 \times 128$ resolution ImageNet dataset: ImageWoof subset in Figure 7 and ImageMeow subset in Figure 8. In addition, parts of the visualizations of synthetic images from the $32 \times 32$ resolution CIFAR-100 dataset are showed in Figure 6. We observe that the synthetic images generated by SeqMatch in the subsequent subset contains more abstract features than the previous subset.

## A.3 Hyperparameter Details

The hyperparameters $K$ of SeqMatch-MTT is set with $\{2, 3\}$ for the settings $\texttt{ipc} = \{10, 50\}$, respectively. The optimal value of hyperparameter $K$ is obtained via grid searches within the set $\{2, 3, 4, 5, 6\}$ in a validation set within the CIFAR-10 dataset. We find that the subset with a small size will fail to condense the adequate knowledge from the corresponding segment of teacher trajectories,

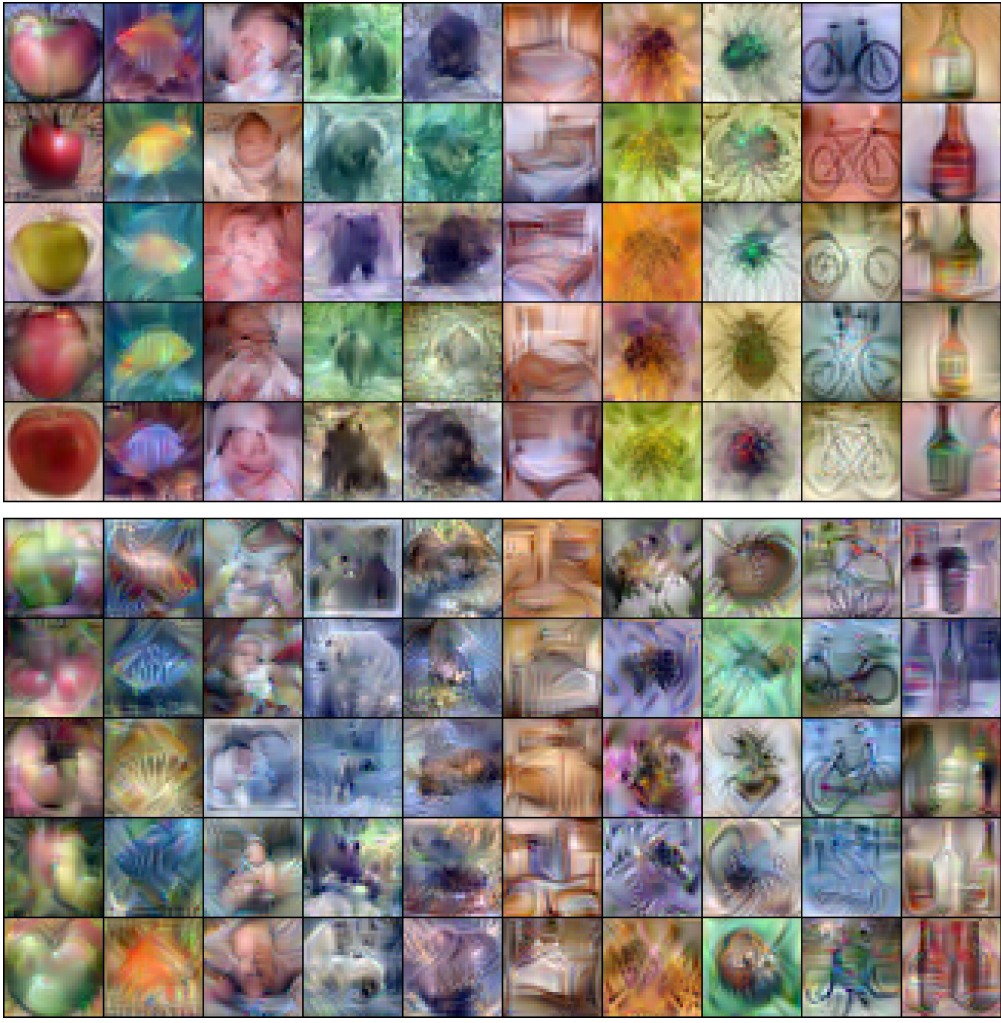

Figure 6: Visualization of the first 10 classes of synthetic images distilled by SeqMatch from $32 \times 32$ CIFAR-100 (`ipc` $= 10$). The initial 5 image rows and the final 5 image rows match the first and second subsets, respectively.

resulting in performance degradation in the subsequent subsets. For the rest of the hyperparamters, we report them in Table 4.

---

[1]ImageFruit has different setting of Max Start Epoch from other ImageNet subesets: {10,10}

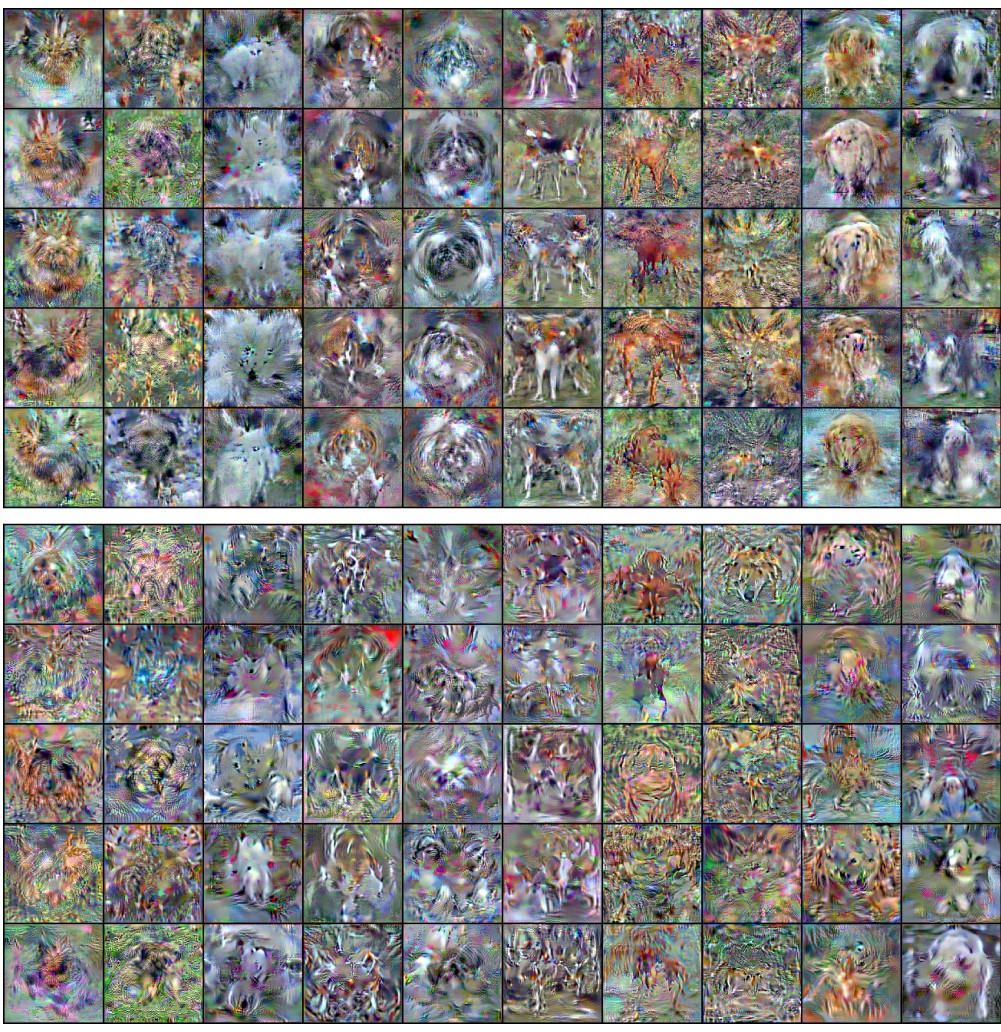

Figure 7: Visualization of the synthetic images distilled by SeqMatch from $32 \times 32$ ImageWoof (`ipc` = 10). The initial 5 image rows and the final 5 image rows match the first and second subsets, respectively.

Table 4: Hyperparameter values we used for SeqMatch-MTT in the main result table. Most of the hyperparameters "Max Start Epoch" and "Synthetic Step" are various with the subsets, we use a sequential numbers to denote the parameters used in the corresponding subsets. "Img." denotes the abbreviation of ImageNet.

| | CIFAR-10 | | CIFAR-100 | | Tiny Img. | Img. Subsets |
|---|---|---|---|---|---|---|
| ipc | 10 | 50 | 10 | 50 | 10 | 10 |
| $K$ | 2 | 3 | 2 | 3 | 2 | 2 |
| Max Start Epoch | {20,10} | {20,20,10} | {20,40} | {40,20,20} | {20,10} | {10,5}[4] |
| Synthetic Step | {30,80} | 30 | 30 | 80 | 20 | 20 |
| Expert Epoch | {2,3} | 2 | 2 | 2 | 2 | 2 |
| Synthetic Batch Size | - | - | - | 125 | 100 | 20 |
| Learning Rate (Pixels) | 100 | 100 | 1000 | 1000 | 10000 | 100000 |
| Learning Rate (Step Size) | 1e-5 | 1e-5 | 1e-5 | 1e-5 | 1e-4 | 1e-6 |
| Learning Rate (Teacher) | 0.001 | 0.01 | 0.01 | 0.01 | 0.01 | 0.01 |

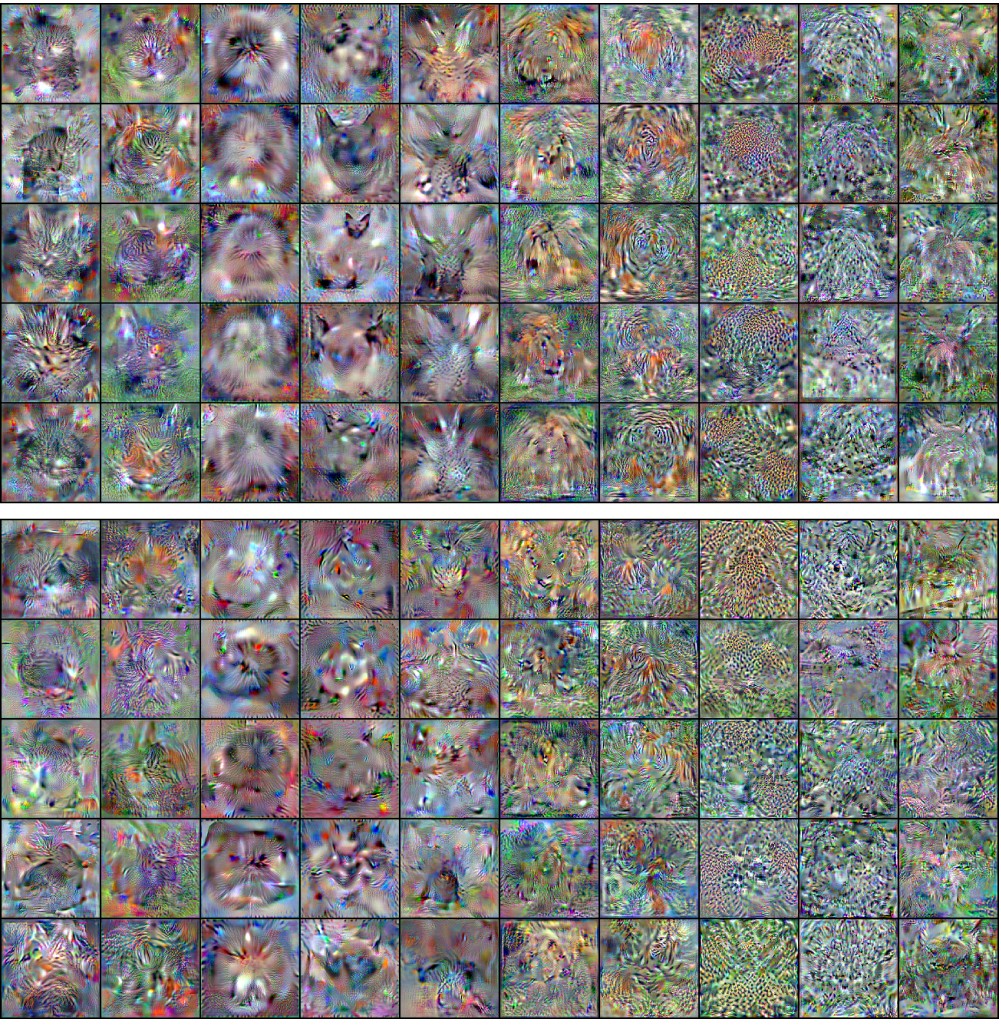

Figure 8: Visualization of the synthetic images distilled by SeqMatch from $32 \times 32$ ImageMeow (`ipc` $= 10$). The initial 5 image rows and the final 5 image rows match the first and second subsets, respectively.

