# OpenReview forum: "Sequential Subset Matching for Dataset Distillation"
_NeurIPS.cc/2023/Conference — NeurIPS 2023 poster_

### Official Review · Reviewer_xCSw · 2023-07-05

**Soundness:** 2 fair
**Presentation:** 3 good
**Contribution:** 3 good
**Rating:** 4
**Confidence:** 4

**Summary:**

This paper proposes a novel dataset distillation method called SeqMatch which focuses on extracting high-level features from later training trajectories. The authors highlight a limitation in state-of-the-art data distillation methods, which tend to condense low-level information from easy data while overlooking the high-level information contained in hard data. In response to this issue, the paper introduces a novel optimization technique that generates multiple small sets of synthetic data. Each of these sets distills distinct knowledge from various stages of the training trajectories. By addressing the inherent problem observed in previous dataset distillation methods, the authors conduct experiments to showcase the efficacy of SeqMatch.

**Strengths:**

1.	The idea of condensing different sub-datasets for different stages is novel and interesting.
2.	The proposed method does not require extra computation cost.
3.	The writing is good and easy to follow.


**Weaknesses:**

-	The paper lacks evaluation numbers for certain settings, such as SeqMatch-IDC, on datasets like CFAIR100 50IPC, Tiny-ImageNet, and ImageNet subset. (SeqMatch-IDC seems to be the most favorable setting) This omission makes it difficult to determine the performance of SeqMatch in comparison to other methods in these specific scenarios.

-	SeqMatch underperforms the baseline method FTD on Tiny-ImageNet and ImageNet subset, suggesting that SeqMatch may not scale well to larger datasets. This raises the question of why SeqMatch was chosen over FTD in these cases.

-	The paper lacks an ablation study, which would provide valuable insights into the impact of the number of subsets, K, on the performance of SeqMatch. Including such an evaluation would enhance our understanding of how SeqMatch operates and how different parameter settings influence its performance.

-	The caption of Table 1 is somewhat misleading. Although IDC is not categorized as a factorization-based method, it does employ data parameterization (factorization can be treated as a special data parameterization). Therefore, it would be more appropriate to compare IDC with RTP and HaBa rather than other methods distilling information into a single image.

-	Minor: There are citation inconsistencies in the appendix. For example, FTD is referred to as [12] in the appendix but as [11] in the main text. Additionally, the appendix lacks a reference section, making it difficult to trace the sources of the cited works accurately.

- The reproductivity checklist is chosen as Yes, but no code is provided.


**Questions:**

How is the repeated evaluation conducted to calculate the error bar?

The motivation of the paper is that SOTA methods fail to distill high-level information from hard data. From my perspective, a more natural and straightforward solution would be to divide the original training dataset into different subsets based on data difficulty and then distill distinct datasets based on these subsets. I am wondering if the authors explored this approach or considered it as an alternative solution in their research.


**Limitations:**

Yes, the authors have discussed the limitations.

---

> ### Author Rebuttal · Authors · 2023-08-10
>
> Thank you for the comments and suggestions. We answer your questions in order.
>
> **Q1:** The paper lacks evaluation numbers for certain settings.
>
> **A1:** This is attributed to the notably sluggish training speed observed in the IDC[21] framework. Our experimental setup aligns with the configurations outlined in IDC[21], and DREAM[51], all of which exclusively present results of IDC under ipc=10, CIFAR 100.
>
> Unlike other gradient-matching baselines, IDC[21] employs an additional class-wise loop asthe complex three-level nested loop structure. This nesting, results in slow training speed, which takes up tp 14 days on a single Nvidia V100 GPU (ipc=50 CIFAR 100).
>
> Addressing your suggestion, we have initiated the training of SeqMatch-IDC with ipc=50, CIFAR100, and we are committed to incorporating the updated results.
>
> **Q2:** underperforms the FTD on Tiny-ImageNet and ImageNet subset.
>
> **A2:**  As asserted in lines 77-78, our SeqMatch serves as a training strategy seamlessly integrable into widely used dataset distillation frameworks. The central thrust of our contribution lies in uncovering a general, yet often unnoticed, limitation within existing distillation methods, wherein each synthetic dataset tends to encapsulate homogeneous features. Concomitantly, we propose an innovative and effective technique to mitigate this particular challenge.
>
> FTD[11] outperforms our base distillation method MTT[5] significantly on Tiny-ImageNet and ImageNet subsets, thereby diminishing the performance improvement attributed to SeqMatch-MTT. In light of this observation, we have initiated experiments utilizing FTD[11] as our foundational distillation approach and are committed to incorporating the updated results into the forthcoming version of our paper.
>
> **Q3:** The paper lacks an ablation study.
>
> **A3:** We appeciate your feedback and have conducted the experiments on the ablation study of K on the number of subsets K on CIFAR10 with SeqMatch-MTT. The results are listed below,
>
> | K |1|2|3|4|5|
> |:----------- | :-----------: |:-----------: |:-----------:|:-----------: |:-----------:|
> |ipc=10|65.3|66.2|65.6|65.0|63.5|
> |ipc=50|71.6|73.2|74.4|74.1|74.3|
>
> Additionally, we present the scatter plot, denoted as **Figure 5** in the provided response PDF, which illustrates the findings from the ablation study conducted on the parameter K. The outcomes distinctly indicate a decline in performance as K escalates from 2 to 5 when ipc=10. A plausible rationale for this phenomenon lies in the fact that subsets with insufficient images (ipc < 5) struggle to effectively distill comprehensive knowledge from the target dataset.
>
> Conversely, the degradation in performance as K increases remains marginal when ipc=50. This can be attributed to the subset size surpassing the threshold required to adequately capture essential features (ipc > 10), thereby substantiating the observed trend.
>
> **Q4:** The caption of Table 1, citation inconsistencies, and the reproductivity checklist.
>
> **A4:** We extend our gratitude for the meticulous review and extend our apologies for the errors identified. Acknowledging the oversight, we concede that the baseline IDC[21] should indeed be classified as a factorization-based method. Consequently, we will diligently revise the presentation in Table 1 within our manuscript.
>
> We are committed to modifying the citation indexing and ensuring proper reference inclusion within both the present and forthcoming iterations of our work. With regard to the availability of our code, we intend to make it publicly accessible upon acceptance of our submission.
>
> **Q5:** How to calculate the error bar?
>
> **A5:** As asserted in lines 283, each network is initialized **5 times** with different random seed to evaluate the synthetic datasets. We follow the experimental setup outlined in the baselines MTT[5] and IDC[21]. We report the mean and variation of the accuracies.
>
> **Q6:** divide the original training dataset  based on data difficulty.
>
> **A6:** We conducted experiments wherein the original dataset was partitioned into three equal subsets termed "Easy," "Medium," and "Hard," respectively. The sorting criterion is based on the instance-wise average loss reduction observed during standard training, following[14]. Subsequently, the synthetic dataset underwent division into three subsets, each earmarked for distillation from the corresponding "Easy," "Medium," and "Hard" target subsets. Post distillation, we proceeded to assess the synthetic dataset through two distinct approaches:
>
> (a) The neural network was trained using the synthetic dataset in the order of "Easy," "Medium," and "Hard", which is marked as "Sequential" in the following table.
>
> (b) The neural network was trained utilizing the entirety of the synthetic dataset, which is marked as "Mixed" in the following table.
>
> We report the results with the reference of our SeqMatch below:
>
> ||Easy |Medium |Hard |Entire|
> |:-:| :-: |:-: |:-:|:-: |
> |Sequential|61.7|65.3|65.9|65.9|
> |Mixed|-|-|-|65.5|
> |SeqMatch|67.9|70.8|74.4|74.4|
>
> The results highlight a significant performance gap between the Divide-and-Conquer approach and our proposed SeqMatch. This discrepancy arises because feature extraction in standard training originates from mini-batches drawn from the entire dataset. The gradients, considered as the teacher trajectories in dataset distillation, constitute a mixture derived from both "Easy" and "Hard" instances.
>
> Despite "Easy" instances with marginal training loss contributing a lower proportion of gradients during subsequent training, these instances play a pivotal role in stabilizing the optimization direction once the network becomes overfitted to the "Hard" instances.
>
> Consequently, dividing the original training dataset along the instance dimension is not a more favorable choice compared to division within the "Epoch" dimension. The latter approach is the methodology implemented in SeqMatch.

---

> > ### Author Response · Authors · 2023-08-17
> > **A request for reviewing the rebuttal**
> >
> >
> > Dear reviewer xCSw，
> > Thank you for your time and dedication to the review process. We are writing to kindly request your response to the rebuttal we submitted in response to your valuable feedback. We understand that reviewers have busy schedules. However, we are eagerly anticipating your constructive suggestions to ensure the effective resolution of any misunderstandings. With only 4 days remaining in the discussion period, we are concerned that there may not be sufficient time to further clarify any misconceptions in our submission.
> >
> > If you could spare a moment to review our rebuttal and share your thoughts, it would immensely help us enhance the quality of the submission. We kindly ask for your comments on any remaining concerns you may have about our submission. Your assistance in this matter would be greatly appreciated.
> >
> > We truly value your input and eagerly look forward to your response at your earliest convenience.
> >
> > Best
> >
> > The authors

---

> > > ### Comment · Reviewer_xCSw · 2023-08-18
> > > **Thanks for the response**
> > >
> > > Thanks for the response. The reason that I replied a bit late is that I am still waiting for the evaluation results promised in A1, but it seems that it is not ready yet.
> > >
> > > I appreciate the effort put into addressing my previous concerns. However, after considering the response and taking into account other reviews, I still have the following concerns: 1. I am waiting for the evaluation results that the authors promised in A1. 2. As mentioned by other reviewers, the improvement is not so significant -- sometimes less than (1%), which hinders the contribution of this paper. 3. I find a misalignment between the stated motivation and the methodology. SeqMatch does not directly address the challenges discussed in the motivation: distilling high-level information from hard data.
> > >
> > > Minor in response: IDC is a data parameterization method but not a factorization-based method.
> > >
> > > Based on the above concerns, I am inclined to keep my rating unchanged.

---

> > > > ### Author Response · Authors · 2023-08-19
> > > > **2nd Response (1/2)**
> > > >
> > > > We appreciate your continued feedback. We anticipate engaging in further discussions and would like to provide the following clarification at this point.
> > > >
> > > > **Q7:**  the evaluation results promised in A1, but it seems that it is not ready yet.
> > > >
> > > > **A7:** The complete results are not ready as the training speed of IDC[21] in cifar100 with ipc=50 is slower than we expected. Because we have to reduce the synthetic batchsize from 64 to 40 due to the huge memory cost caused by the multi-formation technique of IDC[21]. We present the current results at $4713^{th}$ ($6000$ in total) iteration as below,
> > > >
> > > > ||S1 |S2 |S3 |Entire|
> > > > |:-:| :-: |:-: |:-:|:-: |
> > > > |Haba|-|-|-|47.0|
> > > > |IDC|-|-|-|51.4|
> > > > |SeqMatch-IDC|46.1|48.9|51.0 (713/2000)|51.0 (4713/6000)|
> > > >
> > > > Our proposed **SeqMatch-IDC** is currently approaching the baseline IDC with 1287 itertions to go.
> > > >
> > > > ---
> > > >
> > > > We also embed our proposed SeqMatch with the baseline FTD[11]. We evaluate **SeqMatch-FTD** on the two subsets of ImageNet, ImageNette and ImageFruit, we  report the results as below,
> > > >
> > > > | |ImageNette|ImageFruit|
> > > > |:-: | :-: |:-:|
> > > > |MTT[5]|63.0|40.3|
> > > > |Haba[32]|64.7|42.5|
> > > > |FTD[11]|67.7|44.9|
> > > > |SeqMatch-FTD|**70.6** (+2.9)|**46.5** (+1.6)|
> > > >
> > > > SeqMatch-FTD enhances FTD with an average increase of **2.25%** in performance. Notably, SeqMatch-FTD showcases a notable enhancement of **2.90%** on the ImageNette subset, underscoring SeqMatch's superior performance on larger-scale datasets. The ongoing experiments on the ImageWoof and ImageMeow subsets are currently in progress.
> > > >
> > > > ---
> > > >
> > > > **Q8:** the improvement is not so significant
> > > >
> > > > **A8:** MNIST and SVHN are both digit recognition datasets where existing dataset distillation methods could achieve a marginal performance drop after distillation. However, SeqMatch-IDC still achieves a 92.1% accuracy on SVHN with ipc=50, which outperforms other baselines with an accuracy increase of more than **1.90%**.
> > > >
> > > > On the CIFAR datasets, SeqMatch has acquired an **averaged** performance improvement of **2.25%** over the baseline MTT[5] and **0.63%** over IDC[21]. The corresponding **averaged** performance improvement on ImageNet Subsets are **3.88%** over MTT[5] and **2.25%** over IDC[21].
> > > >
> > > > Of greater significance, SeqMatch functions as a training strategy capable of seamless integration into established dataset distillation methods. Our experiments involving the incorporation of SeqMatch into three advanced baseline approaches, namely MTT[5], IDC[21], and FTD[11], serve as validation for the efficacy of our proposed SeqMatch.
> > > >
> > > > ---
> > > > **Q9:** SeqMatch does not directly address the challenges discussed in the motivation: distilling high-level information from hard data.
> > > >
> > > > **A9:**  We begin by demonstrating the **existence of the challenge** through experiments depicted in Figure 1. In these experiments, the synthetic dataset, designed to encapsulate information from the original dataset, exhibits **a failure to reduce the loss associated with hard data** while **overly reducing the loss of easy data**. These outcomes highlight the challenge that current methods encounter in distilling high-level information from hard data.
> > > >
> > > > To further validate the impact of SeqMatch, we observe that it effectively **encourages** baseline methods to prioritize the extraction of high-level information from hard data. This is evident as the **loss of hard data** experiences **a significant reduction** in comparison to the vanilla baseline, as depicted in Figure 1. And the **excessive focus on easy data is mitigated**. Furthermore, the heatmap presented in Figure 4 of the Appendix corroborates the same contribution highlighted in Figure 1.
> > > >
> > > > Hence, we validate that SeqMatch improves the performance of baselines by promoting the distillation of high-level information from challenging data instances.
> > > >
> > > > **We believe our divergence lies in the way to regularize the distillation from hard data.**
> > > >
> > > > The most strightforward approach to do so is **only** distilling information from  hard data subset, which is as you suggested in $1^{st}$ review. However, our response in A6 confirms the **inadequacy of this approach** due to its poor performance. This outcome is logical since the **vanilla training doesn't differentiate** this hard data subset either. The instance sampling remains consistent throughout the entire standard training process. Simply learning or distilling solely from hard data would lead to a severe issue of **catastrophic overfitting** on hard data.
> > > >
> > > > In fact, the **discrepancy across epochs holds a much greater significance** in terms of the density of high-level information compared to **differences at the instance level**. As a result, we opt for regularization in an epoch-wise manner rather than instance-wise. **Each synthetic subset follows distinct learning trajectories across various epochs** (e.g., S1 learns from epoch 0-10, S2 learns from epoch 10-20, and so on). This is also the rationale behind the emphasis on the term "sequential" in naming of our method.

---

> > > > > ### Author Response · Authors · 2023-08-19
> > > > > **2nd response (2/2)**
> > > > >
> > > > > **minor** IDC
> > > > >
> > > > > **Ans** We refer to the survey paper by Lei et al.[28] (Sec. 5, Para. 2, and Table 1), which categorizes IDC[21] as a factorization-based method. The key point of contention lies in whether the multi-formation function should be classified as a decoder. The survey paper by Lei et al.[28] confirms that the multi-formation function indeed functions as a decoder.
> > > > >
> > > > > However, we concur with your viewpoint that IDC[21] should not be classified as a factorization-based method due to the utilization of the training-free bilinear upsampling algorithm for upsampling the partitioned data in the multi-formation function. Consequently, we will incorporate a specific footnote for IDC[21] in Table 1.
> > > > >
> > > > >
> > > > > [28] Shiye Lei and Dacheng Tao. A comprehensive survey to dataset distillation. arXiv preprint arXiv:2301.05603, 2023. 3, 6, 7, 8

---

> > > > ### Author Response · Authors · 2023-08-20
> > > > **Updating of the experiment results**
> > > >
> > > > Dear reviewer xCSw,
> > > >
> > > > We would like to update our results on the ImageNet subsets of SeqMatch-FTD as below,
> > > >
> > > > | |ImageNette|ImageFruit| ImageWoof |ImageMeow|
> > > > |:-: | :-: |:-:|  :-: |:-:|
> > > > |MTT[5]|63.0|40.3|35.8|40.4|
> > > > |Haba[32]|64.7|42.5|38.6|42.9|
> > > > |FTD[11]|67.7|44.9|38.8|43.3|
> > > > |SeqMatch-FTD|**70.6** (+2.9)|**46.5** (+1.6)|**41.1**(+2.3)|**45.4**(+2.1)
> > > >
> > > > Overall, SeqMatch-FTD achieves an average performance enhancement of **2.225%** over FTD[11] across the four ImageNet subsets. The results confirm the **consistent performance** of SeqMatch in **larger-scale datasets**.
> > > >
> > > > We still do not finish the experiments of SeqMatch-IDC on CIFAR100 with ipc=50 due to the complex three-level nested loop structure of IDC (4.2 mins/iteration). We update the results below,
> > > >
> > > > ||S1 |S2 |S3 |Entire|
> > > > |:-:| :-: |:-: |:-:|:-: |
> > > > |Haba[32]|-|-|-|47.0|
> > > > |IDC[21]|-|-|-|51.4|
> > > > |SeqMatch-IDC|46.1|48.9|51.7 (1256/2000)|51.7 (5256/6000)|
> > > >
> > > > Despite having **744 iterations remaining for training**, SeqMatch-IDC has already surpassed IDC[21] by **0.3%**. We are confident that SeqMatch-IDC's performance will **further improve** as the training progresses.
> > > >
> > > > We are eager for the opportunity to address any remaining concerns you might have before the discussion period concludes. It's worth mentioning that **both reviewer QS3g and reviewer u5kd** have found our responses **satisfactory**, leading them to **raise the score** of our paper.
> > > >
> > > > Upon acceptance of our paper, we will release the codes and the distilled datasets. We sincerely appreciate your and other reviewers' dedication and time invested in reviewing our work.
> > > >
> > > > Best
> > > >
> > > > The authors

---

> > > > ### Author Response · Authors · 2023-08-21
> > > > **New findings to answer Q9 and updating of experimental results**
> > > >
> > > > Dear reviewer xCSw,
> > > >
> > > > We would like to provide a final update on SeqMatch-IDC's performance on CIFAR100 with ipc=50, as shown below:
> > > >
> > > > ||S1 |S2 |S3 |Entire|
> > > > |:-:| :-: |:-: |:-:|:-: |
> > > > |Haba[32]|-|-|-|47.0|
> > > > |IDC[21]|-|-|-|51.4|
> > > > |SeqMatch-IDC|46.1|48.9|51.9 (1541/2000)|51.9 (5541/6000)|
> > > >
> > > > SeqMatch-IDC has already surpassed IDC[21] by **0.5%**.
> > > >
> > > > Despite the experiments in Figure 1 of our paper and the clarification of A9 in second respose, **we also conduct another experiments to address your concern on Q9.**
> > > > >**Q9:** SeqMatch does not directly address the challenges discussed in the motivation: distilling high-level information from hard data.
> > > >
> > > > Our experiments verify that the synthetic data within **S3 of SeqMatch-MTT** could be integrated with the original dataset to **enhance the performance of a standard ResNet18 training** on CIFAR10/100 datasets, which is shown below,
> > > >
> > > > |ResNet-18|Original Dataset | Original Dataset + S3 of SeqMatch-MTT|
> > > > |:-:| :-: |:-: |
> > > > |CIFAR10|95.74|96.38 (**+0.64**)|
> > > > |CIFAR100|78.05|78.64 (**+0.59**)|
> > > >
> > > > Drawing inspiration from FocalLoss [55], which assigns lower weight to well-classified examples (easy data) and focuses on a sparse set of challenging examples (hard data), we **incorporate the S3 of SeqMatch-MTT** into a standard ResNet18 training to **strengthen the impact of hard data**. If S3 effectively distills high-level information from hard data, this integrated training should improve performance similarly to how FocalLoss [55] operates. The experimental results **align with our expectations**; S3 enhances the standard training of the original dataset by **0.64%** and **0.59%**.
> > > >
> > > > Furthermore, the experimental outcomes provide **additional evidence that SeqMatch indeed promotes the extraction of high-level information from hard data**. Notably, SeqMatch has the capability to operate in reverse, further amplifying the performance improvement achieved through standard training.
> > > >
> > > >
> > > > In details, the standard training employs basic data augmentation and SGD optimizer(0.05 learning rate, 0.9 momentum, cosine annealing lr, 5e-4 Weight decay) for 200 epochs. The batchsize of original dataset is 128, we randomly sample 13 instances from S3 and integrate with the original mini-batch in each iteration (128 original +13 SeqMatch S3).
> > > >
> > > > The authors
> > > >
> > > > [55] Lin, Tsung-Yi, et al. "Focal loss for dense object detection." Proceedings of the IEEE international conference on computer vision. 2017.

---

### Official Review · Reviewer_QS3g · 2023-07-05

**Soundness:** 4 excellent
**Presentation:** 4 excellent
**Contribution:** 4 excellent
**Rating:** 8
**Confidence:** 5

**Summary:**

This work proposes a change to existing dataset distillation methods by sequentially optimizing different subsets at a time. At each iteration, the existing subset is frozen and a new subset of data is *added* to it and optimized. This method allows different subsets of the synthetic data to capture different levels of features required by a network to learn during training time. This method boosts the state of the art for subsets with >1 IPC (since this method does not make sense with 1 IPC).

**Strengths:**

I like this paper a lot. The authors addressed an obvious problem with existing dataset algorithms (all samples capture the same level of features) in an elegant way that is adaptable to all existing and future backbone methods. The authors both empirically and theoretically show that jointly optimizing the entire set couples the gradients in a way that prevents the synthetic set from learning the necessary variety of features.

The visuals are all very nice, clearly illustrating the authors' points.

**Weaknesses:**

Algorithm 1 is a bit confusing to read. According to line 4, it seems that a single network initialization is used to optimize the entire subset (but this can't be true since it would catastrophically overfit). It is also unclear what the $n$ parameter is. As the algorithm is currently presented, I cannot see how MTT can be slotted into it.

Maybe it would be clearer if an additional inner loop was included along with a generic distillation loss and doing away with the sum over $m$?

**Questions:**

I would like for the authors to make Algorithm 1 more clear as described above.

It would also be nice to see visuals when there is just 1 sample per subset (i.e., K=IPC) even with just IPC 2 or 3 since this case would likely have the largest differences between the subsets.

Lastly, I recognize his surname is quite long, but Figure 1 should probably cite "Cazenavette et al." rather than "George et al." :)

**Limitations:**

yes

---

> ### Author Rebuttal · Authors · 2023-08-10
>
> Thank you for your constructive comments! We sincerely appreciate the time and effort you dedicated to reviewing our work and answer your questions as below.
>
> ---
> **Q1:** Algorithm 1 is a bit confusing to read.
>
> **A1:** We have made revisions to our Algorithm 1, and have appended the revised version in the response PDF. Regarding your concern about clarity in the optimization process, we have now included an inner loop in Algorithm 1, which explicitly illustrates how each subset is optimized in SeqMatch. We hope that this addition clarifies the approach and enhances the understanding for readers. We have added a conditional statement (lines 6-9) in Algorithm 1 to provide a clear indication of this step.
>
> In addition, we have revised the section on input parameters to better explain the notation used. Specifically, we have explicitly defined $N$ as the number of iterations in optimizing each subset. We have introduced a new input parameter, "Base Distillation Method" $\mathcal{A}$, which indicates for the embedding of other distillation methods with SeqMatch.
>
> **Q2:**  visuals with ipc = 2 and 3.
>
> **A2:** We have conducted experiments for $K=2$ and $K=3$ subsets with $\texttt{ipc}=1$ in each subsets by our proposed SeqMatch. The experiments is based on the MTT baseline in CIFAR10 dataset. The evaluation accuracies are listed as below,
>
> | IPC|2|3|
> |:----------- | :-----------: |:-----------: |
> |MTT|51.6|54.5|
> |SeqMatch|52.9|57.0|
>
> SeqMatch outperforms MTT with performance enhancemanents of $\\{1.3\\%,2.5\\%\\}$ under the settings ipc=$\{2,3\}$. For SeqMatch, we designated parameter values of max start epoch as ${2, 4, 6}$ while retaining the remaining parameters consistent with the ipc=$10$ configuration. It is noteworthy that SeqMatch(ipc=2) consists of the first two subsets of SeqMatch(ipc=3). As a result,  the visualization corresponding to SeqMatch(ipc=2) is seamlessly embedded as the first two rows within the SeqMatch(ipc=3) visualization.
>
> The visualization is demonstrated as the **Figure 4** in the response PDF.  However, it is imperative to acknowledge that the disparity between MTT and SeqMatch-MTT is comparatively less pronounced than that illustrated in Figure 3 of our paper. This variance can be attributed to the smaller max start epoch configuration (${2, 4, 6}$) for SeqMatch(ipc=3), a parameter designed to ensure subsequent subsets contribute to enhancing the performance of the initial subset.
>
>
> **Q3:** improper citation.
>
> **A3:** We sincerely apologize for the errors in the citation of MTT[5]. MTT[5] is a crucial baseline method that has greatly influenced our current work, serving as a significant milestone in the dataset distillation task. We have made the necessary revisions in the citation, accurately referencing "Cazenavette et al." in the revised version of our paper. We will keep it in mind in our following work to correctly cite the research papers. We are grateful for pointing out the overlooked mistakes of our submission.

---

> > ### Comment · Reviewer_QS3g · 2023-08-14
> > **Response to Rebuttal**
> >
> > Thank you for answering my questions, I raise my score to an 8.
> >
> > The visuals for K=IPC are particularly interesting; several classes have very stark differences between images.
> >
> > For example, looking at the frogs, the first image can almost be thought of as an "average" frog. The next biggest performance boost can then be gained by "learning about" bright green frogs. After that, the next biggest performance boost can be gained by "learning about" brown frogs.
> >
> > I wonder if this process were repeated for more IPC if we would start to distill samples that resemble the "harder" or "long-tail" samples from the original dataset, like one of the red, yellow, or blue frogs you can see [here](https://knowyourdata-tfds.withgoogle.com/#dataset=cifar10&filters=default_segment.cifar10.label.value:frog).

---

> > > ### Author Response · Authors · 2023-08-15
> > > **Thanks for the comments and raising the score**
> > >
> > > Dear reviewer QS3g,
> > >
> > > Thank you for your prompt response! We have taken note of the color variations among synthetic instances from various sequential subsets. Drawing from your insights, we would like to propose a hypothesis: could enhancing the "mutual orthogonality" of the synthetic dataset potentially contribute to improved overall performance? This notion stems from considering each synthetic instance as a fundamental "basis" element.
> > >
> > > In an effort to substantiate this hypothesis, we are contemplating extending the application of SeqMatch to scenarios involving larger ipc values (>50) and a higher number of subsets (K). Such an expansion could serve as a means to test the hypothesis. Your feedback has prompted us to embark on this exploration.
> > >
> > > Best regards
> > > The authors

---

> ### Author Response · Authors · 2023-08-21
> **New findings inspired by your comments**
>
> Dear reviewer QS3g,
>
> We deeply appreciate your constructive suggestions and thorough review of our work.
>
> Drawing inspiration from your comments,
> >start to distill samples that resemble the "harder" or "long-tail" samples from the original dataset
>
> We hypothesize that the synthetic data within S3 (representing hard data) of SeqMatch-MTT can serve as a condensed subset of challenging instances, capable of being utilized to 'concentrate on a sparse set of challenging examples,' similar to the approach of FocalLoss[55]. This potential utilization could enhance the performance of a standard training procedure using the original dataset. Consequently, we conducted experiments to empirically test this hypothesis.
>
> |ResNet-18|Original Dataset | Original Dataset + S3 of SeqMatch-MTT|
> |:-:| :-: |:-: |
> |CIFAR10|95.74|96.38 (**+0.64**)|
> |CIFAR100|78.05|78.64 (**+0.59**)|
>
>  The experimental results **align with our expectations**; S3 enhances the standard training of the original dataset by **0.64%** and **0.59%**. This investigation could be a potential application of SeqMatch, to operate in reverse, further amplifying the performance improvement achieved through standard training.
>
>  In details, the standard training employs basic data augmentation and SGD optimizer(0.05 learning rate, 0.9 momentum, cosine annealing lr, 5e-4 Weight decay) for 200 epochs. The batchsize of original dataset is 128, we randomly sample 13 instances from S3 and integrate with the original mini-batch in each iteration (128 original +13 SeqMatch S3).
>
>  Best Regards
>
>  The authors
>
> [55] Lin, Tsung-Yi, et al. "Focal loss for dense object detection." Proceedings of the IEEE international conference on computer vision. 2017.

---

### Official Review · Reviewer_Qidh · 2023-07-06

**Soundness:** 1 poor
**Presentation:** 3 good
**Contribution:** 3 good
**Rating:** 5
**Confidence:** 3

**Summary:**

This paper propose a new method called sequential subset matching (SeqMatch) for dataset distillation. The proposed method is designed to continuously generate synthetic images at different training (distillation) iteration. This strategy is inspired by the general mechanism of optimization, which captures characteristics (low-level feature) of easy instances in an early stage, but takes characteristics (i.e., high-level feature) from increasingly difficult instances. SeqMatch was applied to various dataset distillation methods and showed marginal but better performance than the method in which SeqMatch was applied or other baseline methods in various four datasets.

**Strengths:**

1. The analysis about the general mechanism of optimization is so insightful that it deserves a lot of attention in other studies.

2. The plots of several figures provide good support for the arguments in this paper. e.g., Figures 1 and 2 well illustrate the effect of the general mechanism of optimization on dataset distillation and the coupling issue, respectively.

**Weaknesses:**

First of all, I don't understand the motivation behind designing some SeqMatch.

1. How was the claim for the optimization mechanism that captures low-level features in the early stages and high-level features in the later stages verified? Figure 1 seems to have been used to verify this claim, but can hard instances and easy instances represent high-level and low-level features, respectively?

2. I don't understand how the analysis of the general mechanism of optimization was used to design the SeqMatch method. In particular, what is the motivation for applying the SeqMatch method when training f_theta in the evaluation phase?


In addition, in Table 1, most of the performance improvements acquired via SeqMatch are very marginal (~0.2). Performance improvement is not an absolute determining factor in judging the superiority of the proposed method, but it can be used to verify that the method works as claimed in this paper.

**Questions:**

1.  In "Gradient Matching Methods" in Section 3, are the gradients from g_1 to g_M the gradients from different layers of a network? Or gradients from another training iteration?

**Limitations:**

This paper adequately addressed the limitation and promised to solve it in future work.

---

> ### Author Rebuttal · Authors · 2023-08-10
>
> **Q1:** How was the claim for the optimization mechanism that captures low-level features in the early stages and high-level features in the later stages verified?
>
> **A1:** The derivation of the claim has been explicitly presented in lines 161-166 of our paper. In support of this claim, it is crucial to highlight that relevant citations [1], [14], and [44] emphasize that Deep Neural Networks (DNNs) exhibit an optimization pattern where they initially prioritize learning from simpler instances and gradually adapt to more complex ones. It is noteworthy that the term "low-level" features, referring to features primarily extracted by the lower layers, is established in [44].
>
> Our contribution lies not merely in the exposition of this claim, but rather in the extension and validation of these findings within the specific context of dataset distillation. By delving into the task of dataset distillation, we broaden the scope of these existing claims, exploring their applicability and implications in a distinct domain.
>
> In the context of dataset distillation, these designated training instances are repurposed as a validation set to assess the efficacy of knowledge encapsulated within the synthetic dataset. An effective synthetic dataset should manifest an equivalent loss reduction across both easy and hard instances, mirroring the behavior of a standard vanilla training set.
>
> However, as vividly depicted in Figure 1, the current distillation method falls short in achieving this equilibrium, resulting in disparate loss reductions for easy and hard instances. Conversely, our proposed method showcases demonstrable improvements in mitigating this discrepancy, effectively bridging the gap between the two instance categories.
>
> **Q2:** how the analysis of the general mechanism of optimization was used to design the SeqMatch method. In particular, what is the motivation for applying the SeqMatch method when training f_theta in the evaluation phase?
>
> **A2:** We have succinctly expounded upon the impetus driving the conception of the SeqMatch methodology within the delineated sections of our manuscript, specifically in lines 65-73 and lines 233-238.
>
> In essence, our motivation is inspited by the findings that simply increasing the size of synthetic dataset bears an analogous semblance to the amplification of weights in a singular-layer perceptron configuration, in which only a limited performance improvement could be gained. Recognizing the necessity for a more discerning approach, we advocate for a paradigmatic transition from the unidimensional architecture of a single-layer perceptron to the multi-layered intricacies of a Multilayer Perceptron (MLP) and thus encourage the distilled data to be optimized in a sequential manner. Therefore, SeqMatch requires the distilled data to be learned in a sequential manner as well in evaluation phase.
>
> **Q3:** Performance improvements acquired via SeqMatch are very marginal (~0.2)
>
> **A3:** SeqMatch has acquired an **averaged** performance improvement of **1.28%** over the baseline MTT[5] and **0.58%** over the **SOTA** IDC[21]. In particular, SeqMatch improves the performance significantly in the setting of **ipc=50** with an improvement of **1.9%** over the baseline MTT[5] and and **0.93%** over the **SOTA** IDC[21]. Furthermore, our method does not only improve specific algorithms. Our sequential learning framework can be widely applied to existing data distillation models, enhancing their performance. We also hope the versatility of our approach can be taken into consideration as a contribution.
>
> It would be **unjust** to characterize these performance enhancements as merely marginal, approaching the 0.2% range.
>
> **Q4:** In "Gradient Matching Methods" in Section 3, are the gradients from g_1 to g_M the gradients from different layers of a network? Or gradients from another training iteration?
>
>
> **A4:** This represents the trajectory of gradients from the initial state $\theta_0$ to the converged weights $\theta_M$". Consequently, $g_1$ through $g_m$ denote the gradients targeted for optimization in each iteration of a conventional optimization process.
>
>
> ---
>
> [1] Devansh Arpit, Stanisław Jastrz˛ebski, Nicolas Ballas, David Krueger, Emmanuel Bengio, Maxinder S Kanwal, Tegan Maharaj, Asja Fischer, Aaron Courville, Yoshua Bengio, et al. A closer look at memorization in deep networks. In International conference on machine learning, pages 233–242. PMLR, 2017. 4
>
> [14] Bo Han, Quanming Yao, Xingrui Yu, Gang Niu, Miao Xu, Weihua Hu, Ivor Tsang, and Masashi Sugiyama. Co-teaching: Robust training of deep neural networks with extremely noisy labels. Advances in neural information processing systems, 31, 2018. 4, 8
>
> [44] Matthew D Zeiler and Rob Fergus. Visualizing and understanding convolutional networks. In Computer Vision–ECCV 2014: 13th European Conference, Zurich, Switzerland, September 6-12, 2014, Proceedings, Part I 13, pages 818–833. Springer, 2014. 2, 4

---

> ### Author Response · Authors · 2023-08-17
> **A request for reviewing the rebuttal**
>
> Dear reviewer Qidh,
>
> We appreciate the time you dedicated to reviewing our submission. Recognizing that you may have a busy schedule, particularly when evaluating areas outside your own research field, we kindly request that you consider investing some time to explore more about dataset distillation. This emerging and pivotal machine learning task holds significant importance.
>
> We hope you can engage in a comprehensive review of both the rebuttal we have submitted and the insightful comments provided by Reviewer QS3g, an expert in the field of dataset distillation.
>
> Your constructive feedback is immensely valuable to us, and we sincerely thank you for your reviewing.
>
> Best
>
> The authors

---

> ### Comment · Area_Chair_LGDG · 2023-08-20
>
> Dear Reviewer Qidh,
>
> This is another friendly reminder to acknowledge that you have read the rebuttal and the other reviews. Please also share how they change your view on the paper, if at all. Thanks again for your service!
>
> Best,
>
> AC

---

> > ### Comment · Reviewer_Qidh · 2023-08-22
> >
> > Sorry for the late reply. I'm late reading some of the previous work on dataset distillation. Most of my concerns have been resolved, so I will raise my initial rating.

---

> > > ### Author Response · Authors · 2023-08-22
> > >
> > > Dear reviewer Qidh,
> > >
> > > Thank you for your feedback during the review of our paper. We have incorporated your suggestions and rephrased lines 146-148 in Section 3 to enhance clarity. We have also carried out additional experiments, as detailed in A2 for reviewer QS3g and in A3, A6, A7, A9 for reviewer xCSw, in order to provide **a comprehensive evaluation** of our proposed SeqMatch.
> > >
> > > Thank you once again for your timely response.
> > >
> > > Best
> > >
> > > The authors

---

### Official Review · Reviewer_u5kd · 2023-07-10

**Soundness:** 2 fair
**Presentation:** 3 good
**Contribution:** 2 fair
**Rating:** 5
**Confidence:** 4

**Summary:**

This paper investigates an issue with dataset distillation, where synthesized datasets tend to overly condense low-level features but fail to efficiently incorporate high-level ones. The authors argue that this is due to existing methods treating the synthetic dataset as a unified entity and equally optimizing each instance, leading to a coupling issue as the size of the synthetic dataset increases. To address this problem, they propose a new dataset distillation strategy called Sequential Subset Matching (SeqMatch). SeqMatch divides the synthetic dataset into multiple subsets and optimizes them in sequence, mimicking the learning process from low-level to high-level features. This approach allows each subset of the synthetic dataset to progressively capture more complex, high-level features, reducing the coupling issue and enhancing overall performance.

**Strengths:**

This paper is well-written, well-motivated, and well-organized. The authors provide comprehensive experiments on various datasets such as CIFAR-10, CIFAR-100, TinyImageNet, and subsets of the ImageNet. They provide insightful analysis of the experimental results, discussing the impact of different factors. The authors provide a detailed algorithm description that translates their theoretical insights into practical application.

**Weaknesses:**

1. It's not clear how generalizable these results are to other tasks and datasets. Their experiments are based on specific DNN architectures and datasets, and the proposed method's effectiveness might vary under different conditions.
2. Since the method divides the synthetic dataset into subsets and optimizes them sequentially, there might be a risk of overfitting, especially when the number of subsets is high or when subsets are small.
3. The method proposed requires a series of sequential optimization processes, which could potentially increase the computational cost and time required for training.

**Questions:**

1. How do you account for the randomness in the initialization of the synthetic dataset, and how does it affect the rate of convergence of each synthetic instance?
2. The authors mention that instances sharing a similar initialization within the same class will converge similarly. Could this lead to any form of bias in your findings?
3. The authors identified that current gradient matching methods prioritize easy instances during early epochs. What is an effective mechanism for identifying or quantifying the 'difficult' instances to create a more balanced emphasis?

**Limitations:**

In section 5.4, the authors discussed two limitations of their work. Their openness in acknowledging these limitations adds to the credibility of their research and provides useful guidance for follow-up studies in this field.

---

> ### Author Rebuttal · Authors · 2023-08-10
>
> **Q1:** It's not clear how generalizable these results are to other tasks and datasets.
>
> **A1:** Our chosen experimental framework adheres rigorously to the configurations delineated by a suite of established dataset distillation benchmarks, specifically MTT[5], DM[46], CAFE[42], KIP[34,45], IDC[21], FTD[11], and Haba[31]. This standardized experimental setup encompasses datasets, network architectures, and evaluation metrics, thereby ensuring an equitable and unbiased comparison among various dataset distillation methodologies.
>
> Furthermore, we extended our experimental endeavors to encompass additional cross-architecture generalization assessments, as presented in Table 2. These results verified the **superior cross-architecture generalization** inherent to our proposed SeqMatch method. It's important to highlight that conducting experiments under outlier dataset or architecture settings would yield outcomes devoid of substantive relevance or meaningful insights.
>
> **Q2:** There might be a risk of overfitting.
>
> **A2:** The size of each subset bears an impact on the efficacy of SeqMatch. Our parameter study on the number of subsets ($K$) reveals a significant correlation: subsets with insufficient instances inevitably falter in their ability to encapsulate essential features inherent to the original dataset. Our parameter study on SeqMatch-MTT in CIFAR10 dataset is reported as below,
>
> | K |1|2|3|4|5|
> |:----------- | :-----------: |:-----------: |:-----------:|:-----------: |:-----------:|
> |ipc=10|65.3|66.2|65.6|65.0|63.5|
> |ipc=50|71.6|73.2|74.4|74.1|74.3|
>
> The outcomes distinctly indicate a decline in performance as K escalates from 2 to 5 when ipc=10. A plausible explanation for this trend is rooted in the inherent challenge faced by subsets characterized by an insufficient number of images (ipc < 5) to effectively encapsulate comprehensive knowledge from the target dataset. An evidense to support this explaintion is that the subset with ipc >=10 will no longer exhibit a decline in performance (results of ipc=50).
>
> **Q3:** which could potentially increase the computational cost and time
>
> **A3:** We acknowledge the escalation in computational demands and have addressed this limitation in lines 360-367 of our paper. However, it's important to note that the augment in computations is not directly proportional to the number of subsets. This is attributed to the reduction in training iterations for each synthetic subset, wherein only a segment of teacher trajectories is condensed.
>
> Of greater significance, the computational expenditure does not constitute the principal concern within the dataset distillation endeavor. Highlighting this, the most recent survey paper on dataset distillation [49] underscores the discernible gap in performance between synthetic and original datasets, alongside the substantial memory overhead inherent to dataset distillation, thereby impeding its broader adoption in real-world scenarios. In addition to augmenting performance, our proposed SeqMatch offers an ancillary advantage – a reduction in memory expenses associated with dataset distillation. We report the memory usage of MTT and SeqMatch-MTT on CIFAR10 as below,
> | IPC |10|50|
> |:----------- | :-----------: |:-----------:
> |MTT|10,108 MiB (100%)|34,540 MiB (100%)|
> |SeqMatch-MTT|7,164 MiB (70.8%)|15,252 MiB (44.2%)|
>
> The memory used by SeqMatch is **reduced** significantly in particular in the settings with huge memory usage (ipc=50).
>
>
>
> **Q4:** How do you account for the randomness in the initialization of the synthetic dataset, and how does it affect the rate of convergence of each synthetic instance?
>
> **A4:**
> As **emphasized** within lines 196-202, it is evident that discrepancies in initialization and pre-assigned labels engender divergence in the convergence patterns of synthetic data. To address this, we introduce a novel metric termed the **"amplification function,"** as presented in line 212, to quantitatively measure these disparities. Our empirical investigations, demonstrated in Figure 2, illustrate that such discrepancies give rise to a coupling issue that obstructs the effective condensation of features from instances within the $\mathcal{S}^-$ subset (comprising instances with small amplification values). This investigation and the associated formulation is one of the **major contribution** of our work.
>
> The existing dataset distillation methods such as MTT[5], IDC[21], and CAFE[42] tend to clone a random instance from the original dataset as the initialization of the synthetic dataset, which aggravates the divergence in the convergence patterns of synthetic data.
>
> **Q5:** The authors mention that instances sharing a similar initialization within the same class will converge similarly.
>
> **A5:** We find the logic behind this inquiry somewhat perplexing. Is the reviewer implying that our claim might not be entirely accurate and could introduce a bias into our findings? In response, we assert that this is not the case. Our claim doesn't serve as either an assumption or a theoretical foundation for our findings. Rather, it represents a reasoned explanation, inspired by references [1], [14], and [44], to interpret the experimental outcomes showcased in Figures 1 and 2 of our paper.
>
> **Q6:** What is an effective mechanism for identifying or quantifying the 'difficult' instances?
>
> **A6:** The establishment of an effective approach for identifying or quantifying instances deemed 'difficult' presents a prospective avenue for future exploration. Currently, our approach involves employing the average loss reduction observed during standard training as a metric to discern these 'difficult' instances, in alignment with the methodology expounded in [14]. An adapted mechanism within the context of dataset distillation, capable of quantifying and recalibrating the intrinsic "hardness" of instances, offers a promising direction for intensified and enhanced dataset distillation strategies.

---

> > ### Author Response · Authors · 2023-08-20
> > **Thanks for the comments and raising the score**
> >
> > Dear reviewer u5kd,
> >
> > We would like to thank you for the valuable advice and for raising the score of our paper.
> >
> > We will incorporate your suggestions, especially regarding the effective mechanism that **quantifies and reinforces** the 'difficult' data in the dataset distillation process.
> >
> > Best
> >
> > The author

---

> ### Author Response · Authors · 2023-08-17
> **A request for reviewing the rebuttal**
>
> Dear reviewer u5kd,
>
> Thank you for your advice and questions regarding the review of our paper. Serving as a reviewer greatly contributes to the advancement of deep learning research, and it is a crucial duty for researchers. We understand that reviewers have busy schedules, but we sincerely hope that you can spare a moment to review the rebuttal we have submitted to address your concerns. We kindly request you to thoroughly review our rebuttal along with the comments from the reviewer QS3g.
>
> Once again, we extend our gratitude for your assistance and hope that we can address all of your questions before the end of the discussion period.
>
> Best
> The authors

---

> > ### Comment · Reviewer_u5kd · 2023-08-20
> >
> > Thanks for addressing my concerns and I appreciate the authors' thorough rebuttal. I would like to raise the score to be 5: Borderline accept.

---

> ### Author Response · Authors · 2023-08-21
>
> Dear review u5kd,
>
> We express our gratitude for your valuable feedback and your dedicated service as a reviewer within the research community.
>
> Best
>
> The authors

---

### Author Rebuttal · Authors · 2023-08-10

We would like to express our gratitude to all the reviewers and the Associate Chair for dedicating their time and effort to the review of our work. We appreciate the constructive questions and suggestions that have contributed to the enhancement of SeqMatch. Based on the feedback received, we have made revisions to our work as outlined below:

1. We have revised Algorithm 1 as per the suggestion made by reviewer QS3g in order to enhance its clarity. We introduced an inner loop to provide explicit illustration of the optimization process for each subset within SeqMatch. Additionally, we incorporated an input parameter labeled "Base Distillation Method" to signify the potential integration of other distillation methods with SeqMatch. The updated version of Algorithm 1 has been appended to the response PDF for reference.

2. We have appended the visulizations of SeqMatch (**Figure 4** in the response PDF)  under the setting ipc=$\{2,3\}$, alongside SeqMatch-MTT, for the CIFAR 10 dataset. The corresponding evaluation accuracies are presented below:

    | IPC|2|3|
    |:----------- | :-----------: |:-----------: |
    |MTT|51.6|54.5|
    |SeqMatch|52.9|57.0|


    SeqMatch outperforms MTT with performance enhancemanents of $\\{1.3\\%,2.5\\%\\}$ under the settings ipc=$\{2,3\}$.

3. We have presented the abation study over number of subsets $K$ as suggested by the reviewer xCSw. The results is visulized as **Figure 5** in the response PDF. The outcomes distinctly indicate a decline in performance as K escalates from 2 to 5 when ipc=10. A plausible rationale for this phenomenon lies in the fact that subsets with insufficient images (ipc < 5) struggle to effectively distill comprehensive knowledge from the target dataset.

    Conversely, the degradation in performance as K increases remains marginal when ipc=50. This is potentially attributed to the subset size surpassing the threshold required to adequately capture essential features (ipc > 10), thereby substantiating the observed trend.

4. We have studies the memory usage of SeqMatch. The high memory overhead is one of the critial bottolneck of dataset distillation [8][49]. In addition to augmenting performance, our proposed SeqMatch offers an ancillary advantage – a reduction (up to **56% reduction**) in memory expenses associated with dataset distillation. We report the memory usage of MTT and SeqMatch-MTT as **Table 3** in the response PDF.

5. We have modified the improper citation and the consistence issue of references. We also append more reference related to our work as below.

[49] Yu, Ruonan, Songhua Liu, and Xinchao Wang. "Dataset distillation: A comprehensive review." arXiv preprint arXiv:2301.07014 (2023).

[50] Cazenavette, George, et al. "Generalizing Dataset Distillation via Deep Generative Prior." Proceedings of the IEEE/CVF Conference on Computer Vision and Pattern Recognition. 2023.

[51] Liu, Yanqing, et al. "DREAM: Efficient Dataset Distillation by Representative Matching." arXiv preprint arXiv:2302.14416 (2023).

[52] Sachdeva, Noveen, and Julian McAuley. "Data distillation: A survey." arXiv preprint arXiv:2301.04272 (2023).

[53] Zhang, Lei, et al. "Accelerating dataset distillation via model augmentation." Proceedings of the IEEE/CVF Conference on Computer Vision and Pattern Recognition. 2023.

[54] Loo, Noel, et al. "Dataset Distillation with Convexified Implicit Gradients." (2023).

---

### Decision · Program_Chairs · 2023-09-21

**Decision:**

Accept (poster)

**Comment:**

The paper presents a plausible approach to dataset distillation. However, the proposed approach seems to introduce rather marginal improvements upon baselines. While this is a potentially critical issue, we do acknowledge that at least under certain settings the method enables certain gains (and reduction in memory consumption). The paper is definitely borderline, but we decided that the merits (namely the insights and inspirations for future work) outweigh the demerits. We recommend accepting the paper. We strongly urge the authors to include the additional discussion and results in the final version.